

# Probabilistic deconstruction
# of a theory of gravity, part I: Flat space

**Sunok Josephine Suh**

Kavli Institute for Theoretical Physics, University of California,
Santa Barbara, CA 93106-4030, U.S.A.

sjsuh@kitp.ucsb.edu

## Abstract

We define and analyze a stochastic process in anti-de Sitter Jackiw-Teitelboim gravity, induced by the quantum dynamics of the boundary and whose random variable takes values in $AdS_2$. With the boundary in a thermal state and for appropriate parameters, we take the asymptotic limit of the quantum process at short time scales and flat space, and show associated classical joint distributions have the Markov property. We find that Einstein's equations of the theory, sans the cosmological constant term, arise in the semi-classical limit of the quantum evolution of probability under the asymptotic process. In particular, in flat Jackiw-Teitelboim gravity, the area of compactified space solved for by Einstein's equations can be identified as a probability density evolving under the Markovian process.



# 1  Introduction

Jackiw-Teitelboim (JT) gravity is a simple model of gravity in $(1 + 1)$-dimensions [1, 2], in which we can consider the spacetime as factorized into a rigid two-dimensional space and a compactified sphere at each point—see Fig 1. Allowing for some minimally coupled matter, we can write the action as

$$I_{\text{JT+matter}}[g, \Phi, \chi] = \frac{1}{4\pi} \int_{\mathcal{M}} d^2 x \sqrt{-g}\, \Phi(R - 2\Lambda) + \frac{1}{2\pi} \int_{\partial\mathcal{M}} ds \sqrt{-h}\, \Phi K + I_{\text{matter}}[g, \chi], \quad (1)$$

where the dilaton field $\Phi$ represents the area of the sphere at each point. The two-dimensional spacetime is constrained to have constant curvature $R = 2\Lambda$ and is thus locally fixed, while

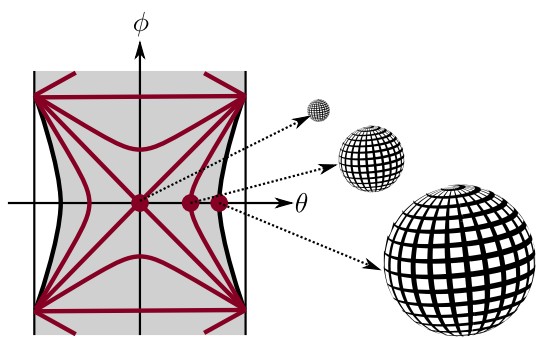

Figure 1: Depiction of the structure of spacetime in JT gravity. The dilaton field, whose level curves are shown, represents the area of the compactified sphere at each point.

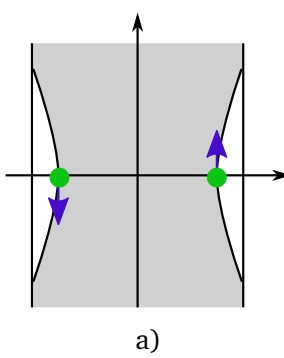
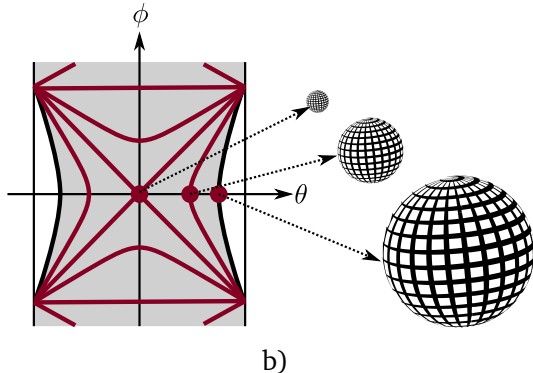

a)                                                    b)

Figure 2: There is a non-trivial holographic duality that maps between a) the total evolution of the boundary particle system in a quantum state close to the thermofield double state and b) corresponding dilaton solution to Einstein's equations, in the limit of large $\gamma$ and low temperature.

Einstein's equations obtained by varying the action with respect to the metric,

$$\frac{1}{2\pi}\left(\nabla_\mu\nabla_\nu - g_{\mu\nu}\nabla^2 - g_{\mu\nu}\Lambda\right)\Phi + T_{\mu\nu} = 0\,, \tag{2}$$

solve for the dilaton. The setting for our discussions will be the anti-de Sitter case of the theory with $\Lambda = -1$, where we have set the radius of curvature to be 1. However, we will also be able to reach conclusions pertaining the flat case of the theory with $\Lambda = 0$, by zooming in to short distances where the spacetime looks flat.

The gravitational theory thus specified is topological—in (1) the bulk action vanishes identically after integrating out $\Phi$, and the only dynamical degrees of freedom are those governed by the boundary action. The boundary action can be characterized as follows: In two dimensions the extrinsic curvature of a curve, up to a total derivative, can be written as an interaction between a spin connection and the unit flow of the curve, $K \cong \pm\omega_\mu\dot{X}^\mu$ (see [3]), so the action is in fact (up to a constant) the world-line action of a particle with spin proportional to $\Phi|_{\partial\mathcal{M}} = \Phi_*$ [4]. Here spin refers to the eigenvalue of quantum wavefunctions of the particle under the generator of local Lorentz transformations. In [4], the theory of this action was regularized in a cutoff-independent manner and quantized so as to yield a canonical ensemble for a single boundary—with each quantum state realized as a particle wavefunction in $\widetilde{AdS_2}$.

In this paper and a sequel [5], the gravitational theory and the quantum theory of its dynamical degrees of freedom described above will be of interest to us as arguably the simplest possible example of holographic duality, or more generally, emergence of a gravitational theory from a quantum theory. Namely, there is a $(0+1)$-dimensional theory describing a boundary particle which can be quantized and solved exactly, which is "dual" to the $(1+1)$-dimensional gravitational theory involving the dilaton field appearing in (1) and (2). (We note the particle theory and its quantum correlators can also be viewed as describing the low-energy dynamics of a microscopic quantum model, the SYK model [6–9].)

Although in this context there is a trivial equality of actions between the two theories, there is still a non-trivial duality to be understood, which captures important semi-classical aspects of what has been studied as the AdS/CFT correspondence [10–12] in general dimensions. In language close to that traditionally used in the literature, there is a mapping between the total evolution of a two-sided boundary particle system in some set of quantum states—excitations around the thermofield double state—and corresponding dilaton solutions to Einstein's equations, in the limit of large coefficient of the boundary action $\gamma = \Phi_*/(2\pi)$ and low temperature, with $\gamma$ larger than inverse temperature. See Figure 2.

A particularly sharp manifestation of this duality and its information-related nature is the ubiquitous Ryu-Takayanagi formula [13,14]: Adapted to our context, it states that the extremal value of the dilaton solution to Einstein's equations equals the von Neumann entropy of the density matrix of one boundary (up to an anomalous term due to the continuum density of states in the boundary theory),[1]

$$\Phi_{\text{ext}} = -\text{Tr}(\rho \ln \rho) + \dots \tag{3}$$

This formula highlights the fact that dilaton configurations in JT gravity, despite not contributing to the action, carry non-trivial information.[2] It also motivates us to ask the following questions: What aspects of the boundary quantum theory are Einstein's equations in JT gravity expressing, such that a formula like (3) could be true? Can we clearly state the significance of general, non-extremal values of the dilaton in the bulk spacetime, and also of the spacetime itself, relative to the quantum theory? And lastly, what kind of expectation values in the quantum theory are relevant to answering these questions?

Effectively, we would like to "deconstruct" gravity in this example, i.e. arrive at a sufficiently new understanding of it such that we can reproduce Einstein's equations from our knowledge of the quantum theory of the boundary. In our investigations, we have found this goal can be achieved by building bridges between quantum theory and notions in probability theory and stochastic processes, the latter crucially having to do with *dynamics* of probabilites.

In this paper, we present a general framework for this deconstruction, and an explicit demonstration that actually pertains to flat JT gravity.[3] Our starting point is to formulate stochastic processes in quantum theory, or *quantum stochastic processes*, using *joint quantum distributions* which are generalizations of joint probability distributions for a classical random variable. Given a quantum system and some observable of the system, joint quantum distributions of the observable are realized as *expectation values of products of projectors* (EVPP's), taking the form

$$q_{T_1}(x_1) = \text{Tr}\big(\rho\, e^{iHT_1} P(x_1) e^{-iHT_1}\big), \quad q_{T_2,T_1}(x_2,x_1) = \text{Tr}\big(\rho\, e^{iHT_1} P(x_1) e^{-iH(T_1-T_2)} P(x_2) e^{-iHT_2}\big), \dots, \tag{4}$$

where $x_i$ are eigenvalues of the operator $X$ corresponding to the observable, and $P(x_i)$ is the projection operator onto the eigenspace of the Hilbert space with eigenvalue $x_i$.

In the setting of JT gravity, we proceed to define a quantum stochastic process in which the observable takes values in $\widetilde{AdS}_2$, using EVPP's in the quantum theory of a single boundary. We note that from the perspective of the boundary theory, $\widetilde{AdS}_2$ is actually a measure space in which a quantum observable—"position of particle"—takes values, and the volume measure of $\widetilde{AdS}_2$ viewed as a spacetime is actually a *probability* measure for the observable which is a manifestation of the Hilbert space of the theory, e.g. it is the measure for integrating the particle wavefunction squared.

In the case that we put the boundary theory in a thermal state, and take the limit of large $\gamma$ and low temperature, we can derive explicit analytic expressions for the EVPP's (4) using results from quantizing the boundary particle obtained in [4]. Furthermore, in the semi-classical limit, which corresponds to taking $\gamma$ to be larger than inverse temperature, we can evaluate integrals involving EVPP's using the saddle-point method. In particular, we can access classical joint probability distributions $p_{T_1}(dx_1), p_{T_2,T_1}(dx_2,dx_1), \dots$, which unlike EVPP's are positive

---

[1]The precise formula is $\Phi_{\text{ext}} = -\text{Tr}(\rho\,(\ln\rho - \ln\text{P}))$ where P is the operator encoding the density of states and enters e.g. a thermal density matrix as $\rho = \int dE\, Z^{-1} e^{-\beta E} \text{P}_{\text{E}}$. See Section 4.2.2 of [4].

[2]One way to view dilaton solutions in JT gravity is as approximations to solutions in a theory slightly perturbed away from JT gravity—for example with a potential $U(\Phi)$ added to the bulk Lagrangian in (1)—which *do* solve a dynamical problem, that of extremizing with respect to $R \neq 2\Lambda$ metric fluctuations.

[3]In most cases of our usage, JT gravity without any qualifications will be referring to anti-de Sitter JT gravity.

everywhere,[4] while accounting for leading saddle-point evaluations of their integrals. Quite generally, we propose that joint probability distributions $p_{T_1}(dx_1)$, $p_{T_2,T_1}(dx_2, dx_1)$, ... arising in this way from integrals of joint quantum distributions $q_{T_1}(dx_1)$, $q_{T_2,T_1}(dx_2, dx_1)$, ... define the classical limit of the quantum stochastic process specified by the latter.

Our discussion thus far applies at all time scales of the dynamics of the boundary system of JT gravity, ranging from short (compared to the radius of curvature of $\widetilde{AdS}_2$, which has been set to 1) to long. When we zoom in to short time scales—and correspondingly, short distances near a point of $\widetilde{AdS}_2$—and extract the behavior of the position observable in *flat space*, we find that the classical stochastic process produced by the quantum stochastic process as described above is in fact a *Markov process*.

We then proceed to consider the *generator equation* of the quantum stochastic process, which characterizes the local (in time) evolution of probability under the process. What we find, working in the same asymptotic, flat limit as before, is that components of the generator equation, at leading non-vanishing order in large $\gamma$ in the semi-classical limit, are in fact Einstein's equations (2) of flat JT gravity, with the dilaton field in the gravitational theory identified as a probability density evolving under the quantum stochastic process! Precisely, we have

$$
\lim_{T_{21}\to 0^+} \int \mathcal{D}x_1 \frac{\partial_{T_2} q_{T_2,T_1}(x_3, x_1)}{q_{T_1}(x_1)} \Phi(x_1)
$$

$$
= \lim_{T_{21}\to 0^+} \lim_{T_{32}\to 0^+} \frac{1}{T_{32}} \Bigg( \int \mathcal{D}x_1 \mathcal{D}x_2 \frac{q_{T_3,T_2,T_1}(x_3, x_2, x_1)}{q_{T_1}(x_1)} \sum_{|\boldsymbol{k}|=0}^{\infty} \frac{\Phi^{(k)}(x_{13}=x_{12})}{\boldsymbol{k}!}(\boldsymbol{x_2}-\boldsymbol{x_3})^k
$$

$$
- \int \mathcal{D}x_1 \frac{q_{T_2,T_1}(x_3, x_1)}{q_{T_1}(x_1)} \Phi(x_1) \Bigg)
\tag{5}
$$

$$
\underset{\text{semi-classical limit}}{\Longrightarrow} \qquad \lim_{x_1 \to x_3} \frac{\partial X^\mu}{\partial l} \frac{\partial X^\nu}{\partial l} \big( \nabla_\mu \nabla_\nu - g_{\mu\nu} \nabla^2 \big) \Phi(x_3) = 0,
\tag{6}
$$

where $l(x_3; x_1)$ is an appropriately scaled geodesic distance from $x_1$ to $x_3$. We note (5) expresses the time evolution of the conditional quantum distribution $q_{T_2,T_1}(x_3, x_1)/q_{T_1}(x_1)$, and is the quantum completion of Einstein's equations—expanding it to higher order in the saddle-point expansion, we will obtain equations at each order in the semi-classical expansion in large $\gamma$, involving increasingly higher derivatives of $\Phi$.

We are thus led to propose the following: JT gravity is in fact a theory concerning a quantum stochastic process, where an observable takes values in a two-dimensional measure space. The dilaton or *area of compactified space* solved for by Einstein's equations is a *probability density*[5] evolving under the process. In the flat case of the theory, the measure space is flat and the process is Markov in the classical limit. The anti-de Sitter case differs foremost in that we have access to a boundary quantum system which induces the relevant quantum stochastic process.[6] Einstein's equations including the cosmological constant term can be derived using the full propagator of the boundary of anti-de Sitter JT gravity rather than its flat asymptotics. This derivation and a precise characterization of the associated stochastic process will be given in [5].

---

[4]As is manifest in EVPP expressions in (4), a single-event quantum distribution $q_{T_1}(dx_1)$ is always positive, while multi-event quantum distributions are in general complex, being neither positive nor real.

[5]Note that we use the term probability even when a measure or measure times density does not integrate to one. See Section 3 for an explanation of our usage.

[6]In classical probability theory, a stochastic process may be induced by a dynamical system of matter, but is intrinsically defined without reference to any such system; see Section 3. We are proposing to define a quantum stochastic process in a similarly self-contained manner, in terms of joint quantum distributions that integrate to one and satisfy marginalization relations.

Extrapolating our analysis of JT gravity, it is natural to conjecture the following. General relativity arises in the semi-classical limit of the evolution of probability with respect to quantum stochastic processes, with the volume measure of spacetime being a probability measure in the target space of a quantum observable. In general, the probability measure will both evolve under and enter the stochastic process governing the observable, and therefore satisfy a non-linear generator equation expressing the time evolution of conditional quantum distributions. (JT gravity is a special case in which the equation is linear because the spacetime factorizes into a base and compactified space, with the volume measure of the base non-fluctuating.) In the leading non-vanishing order in the semi-classical limit, the generator equation has components which are in fact Einstein's equations.

Sections in the rest of the paper are organized as follows: In Section 2, we highlight as well as review and discuss aspects of JT gravity that are relevant to our main analysis. In Section 3, we introduce some notions from probability theory which play a crucial role in our analysis of JT gravity; it is aimed at the reader with a high-energy background. In Section 4 we present our main analysis outlined in the previous paragraphs. In Section 5, we present conclusions, related discussion, and an outline of future research directions.

## 2 Aspects of JT gravity

Here we highlight certain features of Einstein's equations in JT gravity which motivated us to relate its dilaton solution to a probability density associated with the boundary. We also review aspects of the quantum theory of the boundary [3, 4] and discuss the evaluation of EVPP's in the theory.

### 2.1 Einstein's equations

Let us start by writing the action for anti-de Sitter JT gravity [15–17], with a counterterm added to the boundary relative to (1) (and again, having set the AdS radius to 1):

$$I_{\text{JT+matter}}[g, \Phi, \chi] = \frac{1}{4\pi} \int_{\mathcal{M}} d^2x \sqrt{-g}\, \Phi(R+2) + \frac{1}{2\pi} \int_{\partial\mathcal{M}} ds \sqrt{-h}\, \Phi(K-1)$$
$$+ \int_{\mathcal{M}} d^2x \sqrt{-g}\, \mathcal{L}_{\text{matter}}[g, \chi]. \tag{7}$$

The natural boundary condition for the dilaton is Dirichlet: We set $\Phi|_{\partial\mathcal{M}} = \Phi_*$. With this condition and the boundary term involving extrinsic curvature $K$, the variational problem with respect to the bulk metric is well-defined. Since the two-dimensional spacetime $\mathcal{M}$ is constrained to have constant curvature $R = -2$, we can embed $\mathcal{M}$ in $\widetilde{AdS}_2$ and solve Einstein's equations there. As we will see below, the boundary counterterm we have added in (7) is natural in that it renormalizes the energy density of the boundary so that it vanishes (in the absence of any matter) when the boundary coincides with the boundary of $\widetilde{AdS}_2$.

The variations of (7) with respect to the bulk and boundary metric, respectively, are

$$\frac{-2}{\sqrt{-g}} \frac{\delta I}{\delta g^{\mu\nu}} = \frac{1}{2\pi} \left( \nabla_\mu \nabla_\nu - g_{\mu\nu} \nabla^2 + g_{\mu\nu} \right) \Phi + T_{\mu\nu}, \tag{8}$$

$$\frac{-2}{\sqrt{-h}} \frac{\delta I}{\delta h^{ss}} = \frac{1}{2\pi} \left( \Phi_* - N \cdot \partial \Phi \right), \tag{9}$$

where $N$ is the unit normal vector of the boundary with respect to which the extrinsic curvature $K$ is defined, $\dot{X}^\nu \nabla_\nu \dot{X}^\mu = K N^\mu$ where $\dot{X}$ is a unit tangent vector.[7] In the following, we will

---

[7]This normal vector $N$ is the "outward" one with respect to connected components of $\mathcal{M}$.

consider *arbitrary* level curves of a dilaton solution to the Einstein's equations

$$\frac{1}{2\pi}\left(\nabla_\mu \nabla_\nu - g_{\mu\nu}\nabla^2 + g_{\mu\nu}\right)\Phi + T_{\mu\nu} = 0\,. \tag{10}$$

In doing so, we will prefer to use a locally-defined normal vector $n$ obtained by clock-wise rotation of $\dot{X}$, and corresponding extrinsic curvature $\kappa$.

Assuming conservation of the energy-momentum tensor $T_{\mu\nu}$ (which holds when the matter configuration is classical, i.e. solves equations of motion), the equations (10) solved on $\widetilde{AdS}_2$ can be reduced to equations for level curves of $\Phi$, on which $\Phi = \Phi_c$ for some constant $\Phi_c$. They can be put into the form

$$\Phi_c - n \cdot \partial \Phi = (\kappa - 1)n \cdot \partial \Phi - 2\pi n^\alpha n^\beta T_{\alpha\beta}\,, \tag{11}$$

$$\frac{d}{ds}\left(\Phi_c - n \cdot \partial \Phi\right) = 2\pi \dot{X}^\alpha n^\beta T_{\alpha\beta}\,. \tag{12}$$

Given (9), we can identify the expression $\Phi_c - n \cdot \partial \Phi$ as the energy density of the curve. Furthermore, $\dot{X}^\alpha n^\beta T_{\alpha\beta}$ is the flux of energy, and $n^\alpha n^\beta T_{\alpha\beta}$ the flux of transverse momentum. The term $(\kappa-1)n\cdot\partial\Phi$ in (11) can be interpreted as a tension contribution to the energy density, which vanishes as the level curve goes to the boundary of $\widetilde{AdS}_2$ (where $\kappa = 1$). Finally, we can see from (11) that the effect of the counterterm in (7), or equivalently the term $\Phi_*$ in the energy density (9), is to shift the energy density of the boundary as it goes to the boundary of $\widetilde{AdS}_2$, from $-\Phi_*$ to $0$.

What (11) and (12) show is roughly that e.g. time-like level curves of a dilaton solution, which include as a special case the boundary curve $\Phi = \Phi_*$, coincide with *possible* trajectories of a conserving entity, i.e. the boundary particle. We emphasize this identification does not rely on the symmetries of the vacuum with $T_{\mu\nu} = 0$; not only do time-like level curves of a dilaton solution coincide with possible particle trajectories in the vacuum, they *respond to matter* in the same way. This is a striking feature of the equations (10), which motivated us to try to connect dilaton solutions of Einstein's equations to a probability density associated with the evolution of the boundary particle.

For completeness, we record the $\Phi_* \gg 1$, $K - 1 \ll 1$ limit (called the Schwarzian limit in [4]) of (11) and (12) as applied to $\Phi_c = \Phi_*$, although for our purposes it is important to consider the boundary particle at an arbitrary point of $\widetilde{AdS}_2$, and not just near its boundary:[8]

$$\Phi_* - N \cdot \partial \Phi = \Phi_*(K - 1) - 2\pi N^\alpha N^\beta T_{\alpha\beta}\,, \tag{13}$$

$$\frac{d}{ds}\left(\Phi_*(K - 1) - 2\pi N^\alpha N^\beta T_{\alpha\beta}\right) = 2\pi \dot{X}^\alpha N^\beta T_{\alpha\beta}\,. \tag{14}$$

## 2.2 Quantum theory of the boundary

### 2.2.1 Renormalization and parameters

As explained in the introduction, the starting point for quantizing the dynamical, boundary degrees of freedom of JT gravity (whose action is given in (7)) is to recognize its boundary action as a world-line action of a particle with spin. Specifically, for any curve in two dimensions we may write $\pm K = \kappa = \omega_\mu \dot{X}^\mu + \dot{\alpha}$, where $\omega_\mu = (e_1)_\nu \nabla_\mu (e_0)^\nu$ is the gauge field associated with a frame field $\{e_0, e_1\}$ and $\alpha$ is the angle between $e_0$ and unit tangent vector $\dot{X}$.[9] Then

---

[8]A special case of the following equations with specific matter content have appeared in [16].

[9]We have set $e_1$ to be clock-wise rotated from the time-like $e_0$. The expression for $\kappa$ says the following: $\kappa$, or the rate of change of $\dot{X}$ along the curve, is given by the sum of the rate of change of $e_0$ and the rate of change of angle between $e_0$ and $\dot{X}$. See e.g. Appendix A of [3] for its derivation.

the spin of the particle $\nu$ is given in terms of the coefficient of the boundary action, $\nu = \mp i\gamma$, $\gamma = \frac{\Phi_*}{2\pi}$. In our discussions we will fix $\nu = -i\gamma$.

In order to produce a canonical ensemble of quantum states in the Lorentzian spacetime $\widetilde{AdS}_2$, one proceeds by regularizing in the hyperbolic plane $H^2$ the path integral of single-winding closed curves with some fixed length $L$. Note the parameter $L$ is the inverse temperature of the canonical ensemble. According to our discussion above, the boundary action of JT gravity applied to such a curve is given by $I_b = -\gamma \left( \int dX^\mu \omega_\mu + 2\pi - L \right)$ (we have used a subscript $b$ standing for bare). The path integral is regularized by replacing smooth paths with jagged ones consisting of straight segments of length a certain cutoff $\epsilon$. As shown in Section 3 of [4], this results in a quadratic term in the action so that the regularized action is

$$I[X] = \int_0^\beta d\tau \left( \frac{1}{2} g_{\mu\nu} \dot{X}^\mu \dot{X}^\nu - \gamma \omega_\mu \dot{X}^\mu \right), \tag{15}$$

with some renormalized inverse temperature $\beta$.[10]

Let us discuss the cutoff-independent renormalization scheme that was determined in the same reference. The scheme is relevant to the fact that we'll be able to extract the physics of flat space at short (proper) times $T_b \ll 1$. There are two non-universal, possibly cutoff-dependent parameters that enter the renormalization between the bare thermal partition function defined using $I_b$, and its renormalized counterpart defined using $I$. One is the scaling between $L$ and $\beta$, and the other is an overall scaling between the two partition functions. Importantly, in the limit

$$\gamma \gg 1, \quad L \gg 1, \tag{16}$$

we can take the cutoff $\epsilon$ to satisfy $\gamma^{-1} \ll \epsilon \ll 1$ in which case the two parameters are cutoff-independent. In particular, the renormalized inverse temperature is given by

$$\beta = L/\gamma. \tag{17}$$

The physics of this renormalization scheme is that at short distances (or short time scales) compared to the radius of curvature, typical paths are almost straight i.e. reproduce the physics of flat JT gravity, where there is no curvature and thus no spin coupling in e.g. (15). (In contrast, if one takes $\epsilon \to 0$ or $\epsilon \ll \gamma^{-1}, L$, typical paths are jagged.)

In [4], the limit (16) was called the Schwarzian limit. However, in hindsight this is a slight misnomer, as a Schwarzian action (see e.g. Sections 1, 2 of [4]) describes the dynamics of the boundary particle in the limit (16), but *only* at long time scales $T_b \gg 1$. In particular, for our purposes it will be important to probe the Lorentzian dynamics of the particle at short time scales $T_b \ll 1$ in the same limit, where we emphasize the Schwarzian action is not applicable (but the renormalizaton scheme we have described above is valid). Thus we prefer to call (16) the *holographic limit*, in the sense that it is in this limit we will be able to make connections between the boundary quantum theory and a two-dimensional theory of the bulk, e.g. reproduce Einstein's equations from the quantum theory.

Before discussing the Lorentzian theory, let us note the translation of parameters of the particle theory to the microscopic SYK model:

$$\gamma = \alpha_S N, \quad L = \beta_{\text{SYK}} J, \quad \beta = \frac{\beta_{\text{SYK}} J}{\alpha_S N}. \tag{18}$$

Here $N$ is the number of fermions, $J$ is the coupling in the many-body Hamiltonian, and $\alpha_S$ is an order one numerical coefficient in front of the Schwarzian action; see [9]. We note the

---

[10] The constants appearing in the bare action are cancelled in the renormalized thermal partition function in the scheme we describe below.

holographic limit is the analogue of the limit of large $N$ and large 't Hooft coupling in higher-dimensional examples of AdS/CFT. We also note the time scales we will be probing in the particle theory, $T_b \ll 1$, correspond to ultra-short time scales $T_{SYK} \ll J^{-1}$ in the SYK model (as opposed to short time scales $T_{SYK} \sim J^{-1}$) in the language of [9].

### 2.2.2 Lorentzian theory and EVPP's

The Lorentzian theory in $\widetilde{AdS}_2$ is defined by analytically continuing the renormalized action in (15), $S = \int dT \left( \frac{1}{2} g_{\mu\nu} \dot{X}^\mu \dot{X}^\nu + \gamma \omega_\mu \dot{X}^\mu \right)$. In particular, a complete basis of quantum wavefunctions for the boundary particle is obtained by solving the time-independent Schrödinger equation $H\psi = -\frac{1}{2} \nabla^2 \psi = E'\psi$, where $\nabla_\mu = \partial_\mu + \nu \omega_\mu$ is the covariant derivative acting on spinors with spin $\nu$. The energy $E'$ conjugate to renormalized time $T$ agrees with the fully renormalized energy $E$ up to a scheme-dependent additive constant. Using the scheme in the holographic limit described in the previous section, $E' = E - \frac{\gamma^2}{2} + \frac{1}{8}$. Furthermore, using the relation between the Casimir $Q$ of the isometry group $\widetilde{SL}(2,\mathbb{R})$ and the Laplacian acting on $\nu$-spinors, $-\nabla^2 = Q + \nu^2$,

$$E = \frac{s^2}{2}, \tag{19}$$

where $\lambda = \frac{1}{2} + is$, $s > 0$ is the Casimir eigenvalue of irreps forming the Hilbert space.[11] The inner product defining the Hilbert space is an integral over all of $\widetilde{AdS}_2$, $\langle \psi_1 | \psi_2 \rangle = \int_{\widetilde{AdS}_2} d^2x \sqrt{-g} \, \psi_1^*(x) \psi_2(x)$. See Section 4 in [4] for a complete description of the Hilbert space, $\widetilde{SL}(2,\mathbb{R})$-invariant operators on the Hilbert space, and the trace of such operators.

For our purposes we will need to consider expectations values of products of projectors of the form given in (4), involving projectors $P(x) = |x\rangle\langle x|$ onto a point of $\widetilde{AdS}_2$ and a thermal density matrix $\rho$.[12] Using the trace operation defined in the particle theory which factors out the infinite volume of $\widetilde{SL}(2,\mathbb{R})$ from an integral over the Hilbert space, e.g. the one-event and two-event EVPP's are evaluated as[13]

$$q(x_1) = \frac{1}{2} \text{tr}(\rho \, |x_1\rangle\langle x_1|) = \frac{1}{2} \frac{\langle x_1 | \rho | x_1 \rangle}{\text{vol}(\widetilde{SL}(2,\mathbb{R}))}, \tag{20}$$

$$q_T(x_2, x_1) = \frac{1}{2} \text{tr}\left(\rho \, |x_1\rangle\langle x_1| e^{iHT} |x_2\rangle\langle x_2| e^{-iHT}\right) = \frac{1}{2} \frac{\langle x_2 | e^{-iHT} \rho | x_1 \rangle \langle x_1 | e^{iHT} | x_2 \rangle}{\text{vol}(\widetilde{SL}(2,\mathbb{R}))}. \tag{21}$$

Two $\widetilde{SL}(2,\mathbb{R})$-invariant operators (note such an operator $\Psi$ is diagonal in energy, $\Psi = \int dE \, \Psi_E$) whose matrix elements were obtained in [4] enter these expressions: The operator P encoding the density of states of the system $\rho(E)$, $\rho = \int dE \, Z^{-1} e^{-\beta E} P_E$, $Z = \int dE \, e^{-\beta E} \rho(E)$, and the identity operator I whose matrix elements give the particle propagator in $\widetilde{AdS}_2$. In terms of their matrix elements $\langle x_1 | \Psi | x_2 \rangle \equiv \Psi(x_1; x_2)$ we have

$$\langle x_2 | e^{-iHT} \rho | x_1 \rangle = \int dE \, \frac{e^{-\beta E - iET}}{Z} P_E(x_2; x_1), \quad \langle x_1 | e^{iHT} | x_2 \rangle = \int dE \, e^{iET} I_E(x_1; x_2). \tag{22}$$

Let us discuss these two-point functions more explicitly. As detailed in [4], an $\widetilde{SL}(2,\mathbb{R})$-invariant two-point function $\Psi(x_1; x_2)$ is singular and discontinuous across lightrays emanating from $x_2$ and reflecting from the boundaries of $\widetilde{AdS}_2$, see Figure 3a. In our time scales of

---

[11]Note relative to [4], we have changed notation as $E \to E'$, $E_{Sch} \to E$.

[12]The projectors which integrate to one over the target space $\widetilde{AdS}_2$ are actually $P(dx) = dx \sqrt{-g(x)} |x\rangle\langle x|$. However, since the measure $\mathcal{D}x = dx\sqrt{-g}$ is non-dynamical and factors out of our calculations, we consider EVPP's involving $P(x) = |x\rangle\langle x|$ instead.

[13]The trace operation also contains a factor of $\frac{1}{2}$ having to do with there being two boundaries of $\widetilde{AdS}_2$.

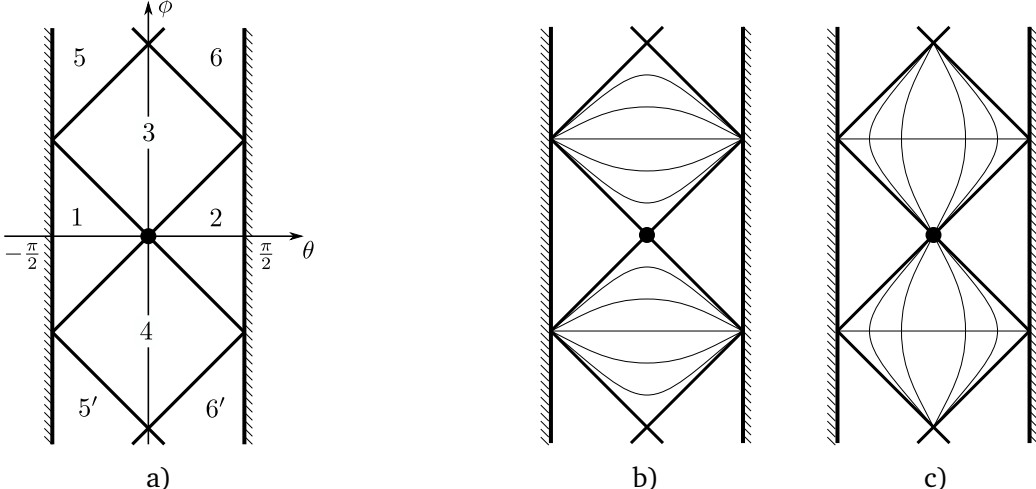

Figure 3: a) Light-rays emanating from a reference point divide $\widetilde{AdS}_2$ into regions. Level curves in our regions of interest $3, 4$, of b) relative coordinate $z$, a function of geodesic distance c) relative Schwarzschild time $t$.

interest $T_b \ll 1$ and in the semi-classical limit, the two-point functions (22) localize to points $x_1$ that are in immediate time-like regions relative to $x_2$, $(x_1; x_2) \in$ regions $3, 4$. Thus we will only need their expressions in regions $3, 4$. Let us introduce coordinates $(\phi, \theta)$ in $\widetilde{AdS}_2$ in terms of which the metric is given by

$$ds^2 = \frac{-d\phi^2 + d\theta^2}{\cos^2\theta}, \qquad -\infty < \phi < \infty, \quad -\frac{\pi}{2} < \theta < \frac{\pi}{2}. \tag{23}$$

For a pair of points $(x; x')$, a cross-ratio measuring geodesic distance can be defined: $w = (\varphi_{13}\varphi_{24})/(\varphi_{14}\varphi_{23})$ where

$$\varphi_{ij} = 2\sin\left(\frac{\varphi_i - \varphi_j}{2}\right), \tag{24}$$

and

$$\varphi_1 = \phi - \theta + \frac{\pi}{2}, \quad \varphi_2 = \phi + \theta - \frac{\pi}{2}, \quad \varphi_3 = \phi' - \theta' + \frac{\pi}{2}, \quad \varphi_4 = \phi' + \theta' - \frac{\pi}{2}. \tag{25}$$

In our calculations in Section 4, we will find it convenient to use as relative coordinates a different cross-ratio $z$ with the property that $0 < z < 1$ in regions $3, 4$, together with a relative Schwarzschild time $t$ having the range $-\infty < t < \infty$ in each region,

$$z = \frac{w}{w-1} = \frac{\varphi_{13}\varphi_{24}}{\varphi_{12}\varphi_{34}}, \qquad t = \frac{1}{2}\ln\left(\left|\frac{\varphi_{14}\varphi_{24}}{\varphi_{13}\varphi_{23}}\right|\right). \tag{26}$$

See Figure 3b,c. Then we note that two-point functions which are matrix elements of operators P and I are given by[14]

$$\mathring{P}_E(x; 0) = \rho(E)(-2)\check{C}_{\lambda,\nu}(w), \quad \mathring{I}_E(x; 0) = (2\pi)^{-2}(-2)\check{C}_{\lambda,\nu}(w), \quad \text{in regions } 3, 4, \tag{27}$$

where

$$\rho(E) = \frac{1}{2\pi^2}\sinh(2\pi s), \tag{28}$$

---

[14]In the "tilde" gauge, a two-point function has the general form $\Psi(x; x') = \left|\frac{\varphi_{23}}{\varphi_{14}}\right|^\nu f_j(w)$, with the radial function $f_j(w)$ coinciding with the two-point function in radial gauge with second point fixed at the origin, $\mathring{\Psi}(x; 0) = f_j(w)$. Here we will not explain choices of gauge, but simply note EVPP's such as (20), (21) are gauge-invariant.

is the density of states in the holographic limit (16), which can be rewritten in the Lorentzian setting as[15]

$$\gamma \gg 1, \qquad s^2 \ll \gamma^2, \tag{29}$$

and $\check{C}_{\lambda,\nu}$ is a function whose exact form for general values of $\gamma$ and $s$ (recall $\nu = -i\gamma$, $\lambda = \frac{1}{2}+is$) is given by

$$\check{C}_{\lambda,\nu}(w) = \lim_{m \to \nu} \frac{\check{A}_{\lambda,m,-\nu}(w) - \check{A}_{\lambda,-m,\nu}(w)}{m - \nu}$$
$$+ \frac{\psi(\lambda + \nu) + \psi(1 - \lambda + \nu) + \psi(\lambda - \nu) + \psi(1 - \lambda - \nu)}{2} \check{A}_{\lambda,\nu,-\nu}(w), \tag{30}$$

where $\check{A}_{\lambda,l,r}(w) = z^{\frac{l+r}{2}}(1-z)^{\frac{-l+r}{2}}\mathbf{F}(\lambda + r, 1 - \lambda + r, 1 + l + r; z)$ with $\mathbf{F}(a,b,c;x)$ the regularized hypergeometric function.

The result (30) was obtained in [4]. At first sight, one may attempt to find a double expansion for it in closed form, first expanding in the holographic limit (29), then in the semi-classical limit

$$s^2 \gg 1. \tag{31}$$

However, this turns out not to be possible. For purposes of this paper, where our goal is to reproduce Einstein's equations in flat JT gravity by extracting the flat-space asymptotics of (30) in the holographic limit, it is actually sufficient to make a cruder approximation where we work at small distances $z \ll 1$ and do not keep full track of $s^2/\gamma^2$ corrections. Furthermore, it is more convenient to work with the differential equation solving for (30),

$$\left(-(1-w)^2(w\partial_w^2 + \partial_w) - \nu^2(1-w)\right)f = \lambda(1-\lambda)f, \tag{32}$$

than with the solution itself. We will present corresponding results in Section 4.2.

Finally, let us comment on infinities appearing in (20), (21). The two-point function $\langle x_1|\rho|x_2\rangle$ is singular as $x_1 \to x_2$, so the numerator $\langle x_1|\rho|x_1\rangle$ in (20) is infinite. However, we can formally evaluate it as follows. Inserting a factor of the identity $\mathbf{1} = \int \mathcal{D}x \, |x\rangle\langle x|$, $\mathcal{D}x = d^2x\sqrt{-g}$ into the equation for unit trace of $\rho$, $1 = \int \mathcal{D}x \frac{1}{2}\mathrm{tr}(\rho \, |x\rangle\langle x|) = \int \mathcal{D}x \frac{1}{2}\langle x|\rho|x\rangle/\mathrm{vol}(\widetilde{SL}(2,\mathbb{R}))$ so

$$\frac{1}{2}\langle x|\rho|x\rangle = \mathrm{vol}(H(x)), \tag{33}$$

where $H(x) \subset \widetilde{SL}(2,\mathbb{R})$ is the isotropy subgroup fixing $x$, of the (left) action of $\widetilde{SL}(2,\mathbb{R})$ on $\widetilde{AdS}_2$. Elements of $H(x')$ fix $x'$ while acting as boosts within each region $1,2,\ldots$ defined by light rays from $x'$, so we may write $\mathrm{vol}(H(x')) = \int_{\text{all regions}} dt$ where $t$ is the Schwarzschild time relative to $x'$ we defined in (26). The infinities $\mathrm{vol}(H)$, $\mathrm{vol}(\widetilde{SL}(2,\mathbb{R}))$ appearing in EVPP's will cancel against infinities coming from integrations over $\widetilde{AdS}_2$ in the integrals we consider in Section 4.

## 3 Notions in probability theory

Here we review some notions in probability theory and stochastic processes. The goal is to introduce the bare minimum needed for understanding our analysis of JT gravity. Much of what we present is a layperson's summary of select material from [18, 19], which we have

---

[15]Note in the (renormalized) Lorentzian theory the bare inverse temperature $L$ does not appear directly, only $\beta$. The condition $L \gg 1$ can be converted to $s^2 \ll \gamma^2$ by using the relation between the proper length and energy of classical curves in $\mathrm{H}^2$, see Section 4.1 of [4].

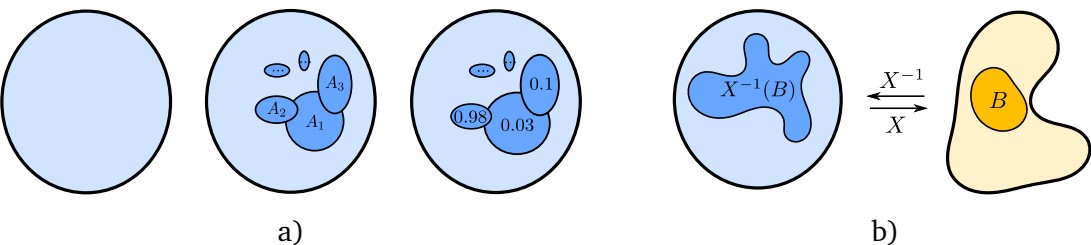

Figure 4: Depiction of a) the three components $(\Omega, \mathcal{F}, P)$ of a probability space, and b) a random variable.

found to be valuable sources of introduction to the subject. Finally, we note the discussion here is entirely concerned with classical probabilities. A crucial step in our task of deconstructing JT gravity will be to make contact with some of the following notions starting from a quantum theory.

**Measure space, probability space:** A set of outcomes $\Omega$ together with an algebra $\mathcal{F}$ of subsets of $\Omega$ constitute a *measurable space* $(\Omega, \mathcal{F})$. A *measure space* $(\Omega, \mathcal{F}, \mu)$ has the additional component of a measure function $\mu : \mathcal{F} \to \mathbb{R}^+$ characterized by additivity on disjoint elements of $\mathcal{F}$. See Figure 4a. In the special case that $\mu(\Omega) = 1$, $\mu$ is called a probability, and $(\Omega, \mathcal{F}, \mu)$ a *probability space*.

Note in our physics discussions we will use the term probability liberally, applying it even to measures that do not integrate to 1 in order to emphasize that they are defined in the context of probability theory. For example, in Section 1 we stated that the volume measure of $\text{AdS}_2$ defined in the context of general relativity is actually a probability measure from the point of view of the quantum theory of the boundary of JT gravity. We will refer to probability measures that do integrate to 1 as probability *distributions*, which is consistent with the definition of the latter given below.

**Random variable, stochastic process:** A *random variable* is a function $X : \Omega \to \Xi$ from a probability space $(\Omega, \mathcal{F}, P)$ to a measurable space $(\Xi, \mathcal{X})$ such that $X^{-1}(B) \in \mathcal{F}$ for each $B \in \mathcal{X}$, i.e. the preimage of a measurable set is measurable. See Figure 4b. It induces a measure on its target space, $\mu_X(B) = P(X^{-1}(B))\ \forall B \in \mathcal{X}$, which is called its *probability distribution*.

A *stochastic process* $\{X_t\}_{t \in T}$ is a collection of random variables $X_t$ indexed by a set $T$, taking values in a common measurable space $(\Xi, \mathcal{X})$. Natural objects characterizing a stochastic process are its finite-dimensional distributions or *joint probability distributions* $\mathbb{P}(X_{t_1} \in B_1, X_{t_2} \in B_2, \ldots X_{t_n} \in B_n)$ for $n \in \mathbb{N}$, $t_1, t_2, \ldots, t_n \in T$, and $B_1, B_2, \ldots, B_n \in \mathcal{X}$. A set of finite-dimensional distributions obeying marginalization relations[16] uniquely specifies a stochastic process, as long as the distributions are sufficiently nice.[17]

We note that a stochastic process may be induced by a dynamical system of matter, but is intrinsically defined without reference to any such system. Explicitly, suppose $\{X_t\}_{t \in T}$ are observables of a dynamical system of matter measured at different times, so that each $X_t : \Omega \to \Xi$ is a function defined on the state space $\Omega$ of the system which has probability function $P$. Then the joint probability distributions of the stochastic process are computed using $\Omega$ and $P$ as

$$\mathbb{P}(X_{t_1} \in B_1, \ldots, X_{t_n} \in B_n) = P(X_{t_1}^{-1}(B_1) \cap \cdots \cap X_{t_n}^{-1}(B_n)). \tag{34}$$

---

[16]That is, $\mathbb{P}(X_{t_1} \in B_1, \ldots, X_{t_n} \in B_n) = \mathbb{P}(X_{t_1} \in B_1, \ldots, X_{t_n} \in B_n, X_{t_{n+1}} \in \Xi, \ldots, X_{t_m} \in \Xi)$ for $m > n$.

[17]The technical requirement is that the finite distributions can be obtained by chaining together conditional probabilities, i.e. $\mathbb{P}(dx_1) = \kappa(dx_1)$, $\mathbb{P}(dx_2, dx_1) = \kappa_2(dx_2; x_1)\kappa(dx_1)$, $\mathbb{P}(dx_3, dx_2, dx_1) = \kappa_3(dx_3; x_2, x_1)\kappa_2(dx_2; x_1)\kappa(dx_1), \ldots$ This can fail if conditional probabilities do not converge appropriately.

However, the definition of the stochastic process only involves the resulting distributions, i.e. we can forget about $\Omega$ and $P$. This will be relevant to our understanding of flat JT gravity vis-à-vis anti-de Sitter JT gravity.

**Probability kernel, Markov operator:** A *probability kernel* from a measurable space $(\Xi, \mathcal{X})$ to another measurable space $(\Upsilon, \mathcal{Y})$ is a map $\mu : \mathcal{Y} \times \Xi \to \mathbb{R}^+$ that captures the notion of conditional probability. In particular, $\forall x \in \Xi$, $\mu(Y; x)$ is a probability measure on $(\Upsilon, \mathcal{Y})$. Two probability kernels are multiplied as $(\mu_2 \mu_1)(dz; x) = \int \mu_2(dz; y) \mu_1(dy; x)$.

It will be important for us that a probability kernel $\mu(dy; x)$ induces a *Markov operator* that maps measures to measures,

$$(M\nu)(dy) = \int \mu(dy; x) \nu(dx). \tag{35}$$

**Markov process, generator:** A stochastic process indexed by one continuous parameter is a *Markov process*, if it models deterministic dynamics where the value of a random variable at a given time determines its distribution at all future times. In particular, a Markov process is fully characterized by its probability kernels $\mu_{t_2, t_1}(dx_2; x_1) = \mathbb{P}(X_{t_2} \in dx_2 | X_{t_1} = x_1)$, which form a semi-group:

1. For all $t$, $\mu_{t,t}(\cdot; x) = \delta_x(\cdot)$. $\tag{36}$
2. For all $t_3 \geq t_2 \geq t_1$, $\mu_{t_3, t_1} = \mu_{t_3, t_2} \mu_{t_2, t_1}$. $\tag{37}$

Let us consider a homogeneous Markov process for which $\mu_{t_2, t_1} = \mu_{t_2 - t_1, 0} \equiv \mu_{t_2 - t_1}$ for all $t_2 \geq t_1$. The Markov operators $M_t$ induced by the probability kernels $\mu_t$ also form a homogenous semi-group, for which a *generator $G$* can be defined which acts on measures $\nu$ in the measurable space of the Markov process,

$$\lim_{t \to 0} \frac{M_t \nu - M_0 \nu}{t} \equiv G\nu. \tag{38}$$

There is a sense in which $M_t$ can be obtained as exponentials of $G$,[18] so the generator obtained as in (38) again fully characterizes the Markov process.

**Generator equation** In our analysis of JT gravity, we will derive Einstein's equations (2) (in the absence of matter) by taking the semi-classical limit of an exact equation expressing the time-derivative of the action of the kernel of a *quantum* stochastic process on a probability measure $\nu(dx) = \mathcal{D}x \, \Phi(x)$. In view of (38), the latter is the quantum version of a *generator equation*.

Let us write down the generator equation for a classical stochastic process with probability kernels $\mu_{t_j, t_i}(dx_j; x_i) = dx_j \mu_{t_j, t_i}(x_j; x_i)$, $t_j \geq t_i$. For an arbitrary measure $\nu(dx) = dx \, \nu(x)$, we have[19]

$$\lim_{t_{21} \to 0^+} \partial_{t_2} (M_{t_2, t_1} \nu)(x_2) = \lim_{t_{21} \to 0^+} \partial_{t_2} \int dx_1 \mu_{t_2, t_1}(x_2; x_1) \nu(x_1)$$

$$= \lim_{t_{21} \to 0^+} \partial_{t_2} \int dx_1 \mu_{t_2, t_1}(x_2; x_1) \sum_{k=0}^{\infty} (x_1 - x_2)^k \frac{\nu^{(k)}(x_2)}{k!}, \tag{39}$$

where in the second line we have used the Taylor expansion of $\nu(x_1)$ about $x_1 = x_2$. For a sufficiently "macroscopic" process for which as $t_2 \to t_1$, $\mu_{t_2, t_1}(x_2; x_1)$ is supported at correspondingly short distances $x_2 \to x_1$, integrals of third-order or higher derivative terms in the

---

[18]See the Hille-Yoshida theorem in e.g. Ch. 12 of [19].

[19]For brevity, we have used notation suitable to the target space of the process being one-dimensional.

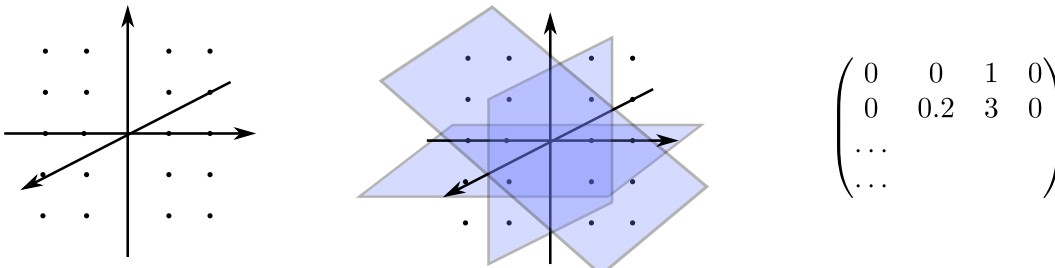

Figure 5: Depiction of the three components $(\mathcal{H}, \mathcal{P}(\mathcal{H}), \rho)$ of a quantum probability space. Compare with those of a classical probability space shown in Figure 4a.

Taylor expansion of $v(x_1)$ are suppressed on the RHS of (39). One is then left with terms involving at most second derivatives of $v$ at $x_2$. The resulting equation is none other than the instantaneous limit of the Fokker-Planck or forward-Kolmogorov equation expressing the time evolution of measures.

As a demonstration, let us derive this generator equation in the simple example of the Wiener process, corresponding to Brownian motion of particles in one dimension. The random variable $X(t) \in \mathbb{R}$ of the Wiener process has independent increments $X(t_2) - X(t_1)$, each of which follow a Gaussian distribution with width $t_{21}$. The probability kernels are given by

$$\mu_{t_2, t_1}(x_2; x_1) = \frac{1}{\sqrt{2\pi(t_2 - t_1)}} e^{-\frac{(x_2 - x_1)^2}{2(t_2 - t_1)}}, \tag{40}$$

and the generator equation (39) reduces to

$$\lim_{t_{21} \to 0^+} \partial_{t_2}(M_{t_2, t_1} v)(x_2) = \lim_{t_{21} \to 0^+} \partial_{t_2}\left(v(x_2) + \frac{t_2 - t_1}{2} v''(x_2) + O\left((t_2 - t_1)^2\right)\right) = \frac{1}{2} v''(x_2). \tag{41}$$

## 4 Stochastic process in JT gravity

### 4.1 Formulation in quantum theory

We begin by noting there are direct analogues in quantum theory of the classical notions of a probability space and a random variable. This was elaborated in Chapter I of [20]; here we explain the bare minimum needed for our purposes, and relegate a more rigorous summary to Appendix A. Essentially, ingredients of a quantum system as known by physicists can be reworked in the language of probability theory; this shift in perspective, although subtle, will be valuable to us in deconstructing JT gravity.

Recall from Section 3 the three components $(\Omega, \mathcal{F}, P)$ of a probability space. A *quantum probability space* is specified by three analogous components $(\mathcal{H}, \mathcal{P}(\mathcal{H}), \rho)$: $\mathcal{H}$ is a Hilbert space, $\mathcal{P}(\mathcal{H})$ is the set of all projections on $\mathcal{H}$ (an operator $T$ is a projection iff $T = T^* = T^2$), and $\rho$ is a density matrix. See Figure 5. We note the probability for an event $E \in \mathcal{P}(\mathcal{H})$ is given by $\text{Tr}(\rho E)$ (cf. the probability for an event $B \in \mathcal{F}$ is $P(B)$). Finally, the analogue of a random variable is an observable, defined as follows: Given a measurable space $(\Xi, \mathcal{X})$, a $\Xi$-valued observable $\xi$ is a projection-valued measure on $(\Xi, \mathcal{X})$, i.e. a mapping $\xi : \mathcal{X} \to \mathcal{P}(\mathcal{H})$ satisfying $\xi(\cup_j F_j) = \sum_j \xi(F_j)$ if $F_i \cap F_j = \emptyset$ for $i \neq j$, and $\xi(\Xi) = \mathbf{1}$.[20]

---

[20] Note the function $\xi$ is actually the direct analogue of the inverse function $X^{-1}$ of a random variable.

When the target space $\Xi$ is a topological space, we can take $\mathcal{X} = \mathcal{X}_\Xi$ to be the algebra generated by the open subsets of its topology. Then we note the definition above is inclusive of—and more general—than the usual physicist's notion of a quantum observable: A self-adjoint operator $T$ is in one-to-one correspondence with a $\mathbb{R}$-valued observable $\xi : \mathcal{X}_\mathbb{R} \to \mathcal{P}(\mathcal{H})$ via $T = \int_\mathbb{R} x\, \xi(dx)$, and for example a bounded[21] normal operator $T$ is in one-to-one correspondence with a bounded $\mathbb{C}$-valued observable $\xi : \mathcal{X}_\mathbb{C} \to \mathcal{P}(\mathcal{H})$ via $T = \int_{\{z | |z| \leq \|T\|\}} z\, \xi(dz)$. (As an aside, we also note the given definition of an observable encompasses both the case of the Hilbert space $\mathcal{H}$ being finite, and of having a countably infinite basis. In the former case, e.g. a self-adjoint operator $X$ corresponds to the observable $\xi(B) = \sum_{x \in B} \xi_x \,\, \forall B \in \mathcal{X}_\mathbb{R}$, where $\xi_x$ is the projector onto the eigenspace $\{u | Xu = xu\} \subset \mathcal{H}$.)

We are now ready to state the significance of the "emergent spacetime" $\widetilde{AdS}_2$ relative to the quantum system of the boundary of JT gravity we described in Section 2.2: It is a measure space

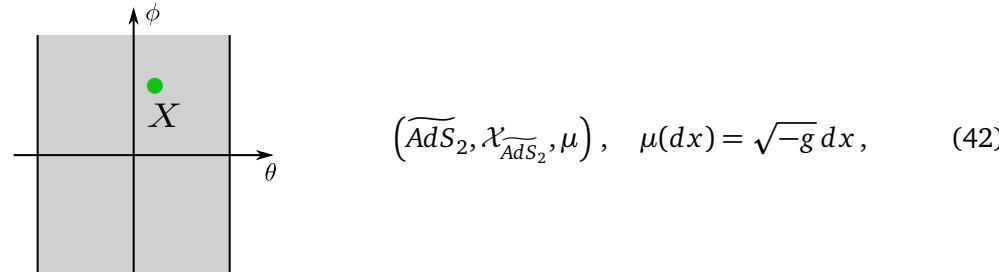

$$\left(\widetilde{AdS}_2, \mathcal{X}_{\widetilde{AdS}_2}, \mu\right), \quad \mu(dx) = \sqrt{-g}\, dx, \qquad (42)$$

in which an observable $X$ of the quantum system—corresponding to "position of particle"—takes values. The measure $\mu$, or the "volume" measure of $\widetilde{AdS}_2$ as viewed in general relativity, is in fact a manifestation of the Hilbert space of the quantum theory which can be characterized as $L_\nu^2(\mu)$— the space of complex-valued functions on $\widetilde{AdS}_2$ transforming as $\nu$-spinors, which are absolutely square-summable under the inner product $\langle f, g \rangle_\mu = \int f^*(x) g(x) \mu(dx)$, $f, g \in L_\nu^2(\mu)$. Stated otherwise, $\mu$ is fundamentally a *probability* measure, e.g. the probability of a particle with wavefunction $\psi$ to be found in some region $B \in \mathcal{X}_{\widetilde{AdS}_2}$ is given by $\int_B \psi^* \psi\, \mu(dx)$. (Formally, this is expressed in the projection-valued measure and operator corresponding to the observable, $\xi(dx) = |x\rangle\langle x| \mu(dx), X = \int_{\widetilde{AdS}_2} x\, \xi(dx) = \int_{\widetilde{AdS}_2} x\, |x\rangle\langle x| \mu(dx)$.)

Our next step is to define a one-parameter stochastic process consisting of the observable $X$ at different proper times. Recall from Section 3 that a classical stochastic process induced by a dynamical system is specified by its joint probability distributions (34) in target space. In our quantum setting, we have the quantum probability space $(\mathcal{H}, \mathcal{P}(\mathcal{H}), \rho)$ of the boundary particle in place of $(\Omega, \mathcal{F}, P)$. Thus we propose to define a stochastic process using *expectation values of products of projectors* or EVPP's[22]

$$\begin{aligned} q_{T_1}(x_1) &= \mathrm{Tr}\!\left(\rho\, e^{iHT_1} |x_1\rangle\langle x_1| e^{-iHT_1}\right), \\ q_{T_2, T_1}(x_2, x_1) &= \mathrm{Tr}\!\left(\rho\, e^{iHT_1} |x_1\rangle\langle x_1| e^{-iH(T_1 - T_2)} |x_2\rangle\langle x_2| e^{-iHT_2}\right), \dots \end{aligned} \qquad (43)$$

The continuum quantum distributions which are analogues of joint probability distributions for the particle to be in range $(x_1, x_1 + dx_1)$ at $T_1, \dots$, and $(x_n, x_n + dx_n)$ at $T_n$ for $n = 1, 2, 3, \dots$ are given by

$$q_{T_n, \dots, T_1}(dx_n, \dots, dx_1) = \mu(dx_n) \dots \mu(dx_1)\, q_{T_n, \dots, T_1}(x_n, \dots, x_1). \qquad (44)$$

---

[21]The norm of an operator $T$ on a Hilbert space $\mathcal{H}$ is given by $\|T\| = \sup_{\|u\|=1} \|Tu\|$ where $u \in \mathcal{H}$.

[22]In the following expression, for simplicity we have assumed the Hamiltonian of the quantum system $H$ is time-independent.

Now, note $q^{(n)} \equiv q_{T_n,\dots,T_1}(dx_n,\dots,dx_1)$ specified as above integrate to 1 by unit trace of density matrix $\rho$ and completeness of projectors $\mathbf{1} = \int \mu(dx)|x\rangle\langle x|$, and also satisfy marginalization relations by virtue of the latter property. However, unlike actual joint probability distributions, $q^{(n)}$ for $n \geq 2$ are in general neither real nor positive ($q^{(1)}$ is always positive). In general, we propose to consider a sequence of distributions $q_{T_n,\dots,T_1}(dx_n,\dots,dx_1)$, $n = 1, 2, 3, \dots$ having these properties to be *joint quantum distributions*, a natural generalization of the notion of joint probability distributions, and to define a *quantum stochastic process* by a set of joint quantum distributions. Then an important question is whether and how contact can be made with a classical stochastic process defined by positive joint distributions.

In [21], what amounts to partial integrals of EVPP's were discussed in the context of attempting to construct a classical version of a sequence of quantum observables. There, and in the literature cited therein, it was imposed that all partial integrals (and thus effectively the EVPP's) must be positive as a precondition of defining an associated classical stochastic process. Here, we take the point of view that this is too stringent of a requirement, and that it is in fact sufficient to be able to extract in the *classical limit*, *effective* joint distributions which are positive and which account for total integrals of joint quantum distributions (unit probability).

Specifically, we propose to extract positive joint distributions $p_{T_2,T_1}(dx_2,dx_1)$, $p_{T_3,T_2,T_1}(dx_3,dx_2,dx_1),\dots$, for ordered times $T_1 \leq T_2 \leq T_3 \leq \cdots$ from leading saddle-point evaluations of total integrals of corresponding joint quantum distributions $q_{T_2,T_1}(dx_2,dx_1)$, $q_{T_3,T_2,T_1}(dx_3,dx_2,dx_1),\dots$:

$$\int_{x_1,\dots,x_n} q_{T_n,\dots T_1}(dx_n,\dots,dx_1) \underset{\text{leading sadd. pt. eval.}}{\approx} \int_{x_1,\dots,x_n} p_{T_n,\dots T_1}(dx_n,\dots,dx_1). \tag{45}$$

The $p^{(n)} \equiv p_{T_n,\dots,T_1}(dx_n,\dots,dx_1)$ defined as such are guaranteed to be positive, essentially because in the leading saddle-point approximation an integral is evaluated along a path of constant phase for the integrand. It is also evident that they integrate to 1 by construction, and inherit marginalization relations. Thus they are bona fide probability distributions which define a classical stochastic process in the target space of $X$. We note this procedure for extracting a classical stochastic process from joint quantum distributions (or EVPP's) is quite general, and can be expected to apply broadly to observables of quantum systems in the classical limit. In the next section, we will apply it in a concrete scenario.

## 4.2 Markov process in local limit

In this section, we formulate conditions stating that classical joint probability distributions produced by a quantum stochastic process—as explained in the previous section—form a Markov process. We then show that the dynamics of the position observable of JT gravity in flat space, which we extract by zooming in near a point of $\widetilde{AdS}_2$ in the holographic limit of the boundary quantum system, produce a Markov process in such a manner.

### 4.2.1 Conditions for a Markov process

We consider a thermal density matrix $\rho$ for the boundary particle. Then our discussion simplifies because $\rho$ commutes with the Hamiltonian and thus the EVPP's are homogeneous in time. Evaluating as in (20), (21), and (33), and using (44) with the more familiar notation

$$\mu(dx) = \sqrt{-g}\, dx \equiv \mathcal{D}x, \tag{46}$$

we have that joint quantum distributions of the boundary observable are given by

$$q(dx_1) = \mathcal{D}x_1 \frac{\text{vol}(H(x_1))}{\text{vol}(\widetilde{\text{SL}}(2,\mathbb{R}))}, \tag{47}$$

$$q_{T_{n,n-1},\ldots,T_{21}}(dx_n,\ldots,dx_1) = \mathcal{D}x_n\cdots\mathcal{D}x_1$$
$$\times \frac{1}{2}\frac{\langle x_n|e^{-iHT_{n,1}}\rho|x_1\rangle\langle x_1|e^{iHT_{21}}|x_2\rangle\ldots\langle x_{n-1}|e^{iHT_{n,n-1}}|x_n\rangle}{\text{vol}(\widetilde{\text{SL}}(2,\mathbb{R}))}\quad (n\geq 2), \tag{48}$$

where $T_{j,k} = T_j - T_k$ are differences in proper times $T_j$ associated with $x_j$. We consider an increasing sequence of times, i.e. $T_{j+1,j} \geq 0$ for $1 \leq j \leq n-1$.

Now, let us consider conditions under which the joint probability distributions $p(dx_1) = q(dx_1)$ and $p_{T_{n,n-1},\ldots,T_{21}}(dx_n,\ldots,dx_1)$ for $n \geq 2$ arising from integrals of (48) via the saddle-point approximation form a Markov process as was defined by (36), (37). In extracting joint probability distributions as such, there is one subtlety specific to our system: Namely the $\frac{1}{2}$ factor appearing in (48) corresponds to taking a quantum superposition of two classically exclusive possibilities of the particle going "up" or "down", and should not enter the definition of joint probability distributions in the classical limit. In other words we have, as slightly modified from the prototype formula (45),

$$\int_{x_1,\ldots,x_n} q_{T_n,\ldots T_1}(dx_n,\ldots,dx_1)\underset{\text{leading sadd. pt. eval.}}{\approx}$$
$$\frac{1}{2}\left(\int_{(x_1,x_2),\ldots,(x_{n-1},x_n)\in\text{region }4}+\int_{(x_1,x_2),\ldots,(x_{n-1},x_n)\in\text{region }3}\right)p_{T_n,\ldots T_1}(dx_n,\ldots,dx_1) \tag{49}$$

(See Figure 3 for a depiction of the pair-wise relative regions 3 and 4). Taking this into account, we extract from the total integral of the two-event quantum distribution[23]

$$\int q_T(dx_2,dx_1) = \frac{1}{2}\int \mathcal{D}x_2\mathcal{D}x_1\frac{\langle x_2|e^{-iHT}\rho|x_1\rangle\langle x_1|e^{iHT}|x_2\rangle}{\text{vol}(\widetilde{\text{SL}}(2,\mathbb{R}))}$$
$$= \frac{1}{2}\int\mathcal{D}z_{21}\langle x_2|e^{-iHT}\rho|x_1\rangle\langle x_1|e^{iHT}|x_2\rangle\underset{\text{leading sadd. pt. eval.}}{\approx}\frac{1}{2}\int\mathcal{D}z_{21}f_T(z_{21}), \tag{50}$$

the probability distribution $p_T(dx_2,dx_1) = \mathcal{D}x_2\mathcal{D}x_1\frac{f_T(z_{21})}{\text{vol}(\widetilde{\text{SL}}(2,\mathbb{R}))}$ and the probability kernel

$$\mu_T(dx_2;x_1) = \frac{p_T(dx_2,x_1)}{p(dx_1)} = \mathcal{D}x_2\frac{f_T(z_{21})}{\text{vol}(H(x_1))}. \tag{51}$$

Recall that the first requirement for a Markov process (36) is that the probability kernel $\mu_T$ induces the identity operator as $T \to 0$. This is verifiable from the explicit form of $f_T$ we will derive in the next section. Here we note that in fact a stronger statement holds, that the exact *quantum* kernel

$$\kappa_T(dx_2;x_1) = \frac{q_T(dx_2,dx_1)}{q(dx_1)} = \mathcal{D}x_2\frac{1}{2}\frac{\langle x_2|e^{-iHT}\rho|x_1\rangle\langle x_1|e^{iHT}|x_2\rangle}{\text{vol}(H(x_1))}, \tag{52}$$

---

[23]In the second equality of (50), we have used $\int\mathcal{D}x = \int\mathcal{D}zdt$ in terms of relative coordinates $(z,t)$ with respect to some arbitrary point. We will use this repeatedly in our evaluation of integrals.

induces the identity operator as $T \to 0$, $\kappa_0(dx_2; x_1) = dx_2\,\delta(x_2 - x_1)$, due to (33). The second requirement (37) corresponding to the actual Markov property can be restated as follows: A joint probability is produced by iterations of the probability kernel,[24]

$$p_{T_{n,n-1},\dots,T_{21}}(x_n, \dots, x_1) = \mu_{T_{n,n-1}}(x_n; x_{n-1}) \dots \mu_{T_{21}}(x_2; x_1) p(x_1).\tag{53}$$

In our quantum context, this translates to the non-trivial condition

$$\int q_{T_{n,n-1},\cdots,T_{21}}(dx_n, \dots, dx_1)$$
$$= \frac{1}{2} \int \mathcal{D}x_n \cdots \mathcal{D}x_3 \mathcal{D}z_{21}\, \langle x_n | e^{-iHT_{n,1}} \rho | x_1 \rangle \langle x_1 | e^{iHT_{21}} | x_2 \rangle \cdots \langle x_{n-1} | e^{iHT_{n,n-1}} | x_n \rangle$$
$$\underset{\text{leading sadd. pt. eval.}}{\approx} \frac{1}{2} \int \mathcal{D}z_{n,n-1} \cdots \mathcal{D}z_{21}\, f_{T_{n,n-1}}(z_{n,n-1}) \cdots f_{T_{21}}(z_{21}), \qquad \text{for } n \geq 3.\tag{54}$$

### 4.2.2 Flat asymptotics

We will now outline how the Markov property as captured by (50), (54) holds for the dynamics of the position observable in flat space, which we can extract from the asymptotics of the observable at short distances of $\widetilde{AdS}_2$ in the holographic limit. (Recall from Section 2.2.1 the holographic limit and associated renormalization scheme make the latter possible.) See Figure 6.

Our starting point is to obtain the $0 < z \ll 1$ asymptotics of the particle propagator (30) in the holographic, semi-classical limit $\gamma \gg s \gg 1$. As we show in Appendix B, this is accomplished by taking the asymptotic limit

$$\gamma \to \infty, \qquad |z| \ll 1,\tag{55}$$

of the equation (32), then using the saddle-point approximation in integral representations for the solutions, which are Whittaker functions $M_{-i\eta,0}(i\xi)$, $W_{-i\eta,0}(i\xi)$ where

$$\eta = \frac{\gamma^2 - s^2}{2\gamma}, \qquad \xi = 2\gamma z.\tag{56}$$

The results are given by

$$\check{C}_{\lambda,\nu}(w) \approx \frac{\sqrt{\pi}}{2(\eta\xi)^{1/4}} \left\{ e^{\frac{3}{4}\pi i} e^{-2i\sqrt{\eta\xi}} \left( 1 + \frac{i}{16} \frac{1}{\sqrt{\eta\xi}} \right) + \text{c.c.} \right\} + O\left( \frac{1}{\gamma^{5/2}} \right).\tag{57}$$

In the limit (55), $(\xi, t)$ are relative coordinates in flat space (cf. $(z, t)$ in $\widetilde{AdS}_2$ given by (26)). In terms of null coordinates of Minkowski space $u = \frac{\phi - \theta}{2}$, $v = \frac{\phi + \theta}{2}$, $ds^2 = -d\phi^2 + d\theta^2$,

$$u - u' \approx \pm\sqrt{|\xi|}e^{-t}, \quad v - v' \approx \pm\sqrt{|\xi|}e^{t},$$
$$\Longleftrightarrow \quad \xi \approx (u - u')(v - v'), \quad t \approx \frac{1}{2}\ln\left| \frac{v - v'}{u - u'} \right|.\tag{58}$$

Finally, we note only the leading terms in $\gamma$ in (57) are relevant to confirming the Markov property of the resulting classical stochastic process. The increased accuracy afforded by the subleading term will be required when reproducing Einstein's equations in the next section.

---

[24]For a non-homogenous process the probability kernels should be labeled by two different times rather than their difference.

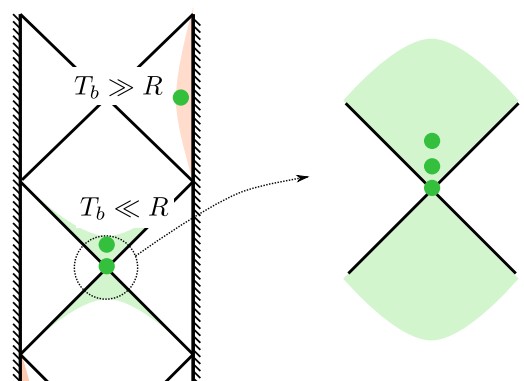

Figure 6: In global AdS$_2$, we zoom in to short times and correspondingly short distances (from a reference point) compared to the AdS radius (green region), to extract the dynamics of the boundary particle in flat space. In contrast, it is at times and distances long compared the AdS radius (red region) that the Schwarzian action applies.

Let us first obtain the two-event kernel $f_T(\xi)$ defined by (50). Using (22), (27), and the semi-classical approximations

$$\rho(E) = \frac{\sinh 2\pi s}{2\pi^2} \approx \frac{e^{2\pi s}}{(2\pi)^2}, \qquad Z = \int dE\, e^{-\beta E} \rho(E) \approx \frac{1}{(2\pi)^2} \left(\frac{2\pi}{\beta}\right)^{\frac{3}{2}} e^{\frac{2\pi^2}{\beta}}, \tag{59}$$

$$\int \mathcal{D}z = 2 \int dz \approx \gamma^{-1} \int_{\text{regions 3,4}} d\xi,$$

we have

$$\frac{1}{2} \int \mathcal{D}z_{21} \, \langle x_2 | e^{-iHT} \rho | x_1 \rangle \langle x_1 | e^{iHT} | x_2 \rangle \tag{60}$$

$$\approx \frac{1}{2} \int_{3,4} \frac{d\xi}{\gamma} \int ds\, s \frac{e^{-(\beta+iT)s^2/2 + 2\pi s}}{\left(\frac{2\pi}{\beta}\right)^{3/2} e^{2\pi^2/\beta}} \frac{\sqrt{\pi}}{(\eta\xi)^{1/4}} \left(e^{-\frac{1}{4}\pi i - 2i\sqrt{\eta\xi}} + \text{c.c.}\right)$$

$$\times \int ds'\, s' \frac{e^{iTs'^2/2}}{(2\pi)^2} \frac{\sqrt{\pi}}{(\eta'\xi)^{1/4}} \left(e^{-\frac{1}{4}\pi i - 2i\sqrt{\eta'\xi}} + \text{c.c.}\right). \tag{61}$$

We proceed to evaluate the integral (60) by saddle-point approximation. For $T = \frac{T_b}{\gamma} > 0$, only the term in (60) with exponential $e^{-p}$,

$$p(s, s', \xi) = (\beta + iT)\frac{s^2}{2} - 2\pi s + 2i\sqrt{\eta\xi} - iT\frac{s'^2}{2} - 2i\sqrt{\eta'\xi}, \tag{62}$$

contributes. The saddle-point is given by

$$s_* = s'_* = \frac{2\pi\gamma}{L}, \qquad \xi_* = \eta_* T_b^2. \tag{63}$$

In order for $\xi_* = 2\gamma z_*$ to agree with the short-time asymptotics of the classical trajectory of the particle obtained from its bare action (see Appendix C), $z_* = \left(\frac{L}{2\pi}\right)^2 \sinh^2\left(\frac{\pi T_b}{L}\right) \approx \frac{1}{4} T_b^2$, we need to take the limit[25]

$$L = \frac{2\pi\gamma}{s_*} \to \infty. \tag{64}$$

---

[25]Technically, this has to do with the fact that we did not retain subleading $s^2$ dependences (in the step of taking limit of differential equation) in deriving the asymptotic form (57).

Thus the propagator (57) is seen to be valid in the *local limit* $z \ll 1$, $L \to \infty$. The local limit captures the notion of "zooming in" near a point, after which the geometry is flat and the infrared cutoff of the spacetime (bare inverse temperature $L$) has moved to infinity. With the understanding that we remove the cutoff as (64) in the final stage of a calculation, after expanding the exponent (62) to quadratic order about its saddle-point and integrating over $s, s'$, we obtain

$$\int q_T(dx_2, dx_1) \approx \frac{1}{2} \int_{\text{regions } 3,4} \gamma^{-1} d\xi \, f_T(\xi), \qquad \gamma^{-1} f_T(\xi) = \frac{1}{\sqrt{2\pi\gamma T_b^3}} e^{-\frac{1}{2}\frac{1}{\gamma T_b^3}\left(\xi - \frac{\gamma}{2} T_b^2\right)^2}. \tag{65}$$

Next, we study the geometry of three consecutively causally-connected points $(x_3, x_2, x_1)$ in flat space, which is needed to verify $n$-point probabilities for $n \geq 3$ decompose as in (54). Specifically, we solve for $\xi_{31}, t_{31}$ as functions of $\xi_{32}, t_{32}, \xi_{21}, t_{21}$ in the limit (55). Only the case $\text{sgn}(\phi_{32})\,\text{sgn}(\phi_{21}) > 0$ will be relevant to saddle-point evaluations of integrals of $n$-point EVPP's:

$$\xi_{31} \approx \xi_{32} + \xi_{21} + 2\sqrt{\xi_{32}\xi_{21}}\cosh(t_{32} - t_{21}),$$

$$t_{31} \approx \frac{1}{2}\ln\left(\frac{\sqrt{\xi_{32}}e^{t_{32}} + \sqrt{\xi_{21}}e^{t_{21}}}{\sqrt{\xi_{32}}e^{-t_{32}} + \sqrt{\xi_{21}}e^{-t_{21}}}\right). \tag{66}$$

Using the asymptotic propagator (57), the associated limit (64), and the geometry of flat space as reflected in (66), it is possible to show the Markov property (54) holds with probability kernel given by (65). We give a detailed proof in Appendix D. In the semi-classical approximation,

$$\frac{1}{2}\int \mathcal{D}x_n \cdots \mathcal{D}x_3 \mathcal{D}z_{21} \langle x_n|e^{-iHT_{n,1}}\rho|x_1\rangle \langle x_1|e^{iHT_{21}}|x_2\rangle \cdots \langle x_{n-1}|e^{iHT_{n,n-1}}|x_n\rangle$$

$$\approx \frac{1}{2}\int_{3,4} \frac{d\xi_{n,n-1}dt_{n,n-1}}{\gamma} \cdots \int_{3,4}\frac{d\xi_{32}dt_{32}}{\gamma}\int_{3,4}\frac{d\xi_{21}}{\gamma}\int ds\, s \frac{e^{-(\beta + iT_{n,1})s^2/2 + 2\pi s}}{\left(\frac{2\pi}{\beta}\right)^{3/2}e^{2\pi^2/\beta}}$$

$$\times \frac{\sqrt{\pi}}{(\eta\xi_{n,1})^{1/4}}\left\{e^{-\frac{1}{4}\pi i - 2i\sqrt{\eta\xi_{n,1}}} + \text{c.c.}\right\}$$

$$\times \prod_{j=1}^{n-1}\int ds_j\, s_j \frac{e^{iT_{j+1,j}s_j^2/2}}{(2\pi)^2}\frac{\sqrt{\pi}}{(\eta_j\xi_{j+1,j})^{1/4}}\left\{e^{-\frac{1}{4}\pi i - 2i\sqrt{\eta_j\xi_{j+1,j}}} + \text{c.c.}\right\}. \tag{67}$$

Applying the saddle-point approximation, for $T_{j+1,j} > 0$, $1 \leq j \leq n-1$ only the phase term $e^{-p}$ with

$$p(\xi_{n,n-1}, t_{n,n-1}, \ldots, \xi_{32}, t_{32}, \xi_{21}, s, s_1, \ldots, s_{n-1})$$

$$= (\beta + iT_{n,1})\frac{s^2}{2} - 2\pi s + 2i\sqrt{\eta\xi_{n,1}} - \sum_{j=1}^{n-1}\left(iT_{j+1,j}\frac{s_j^2}{2} + 2i\sqrt{\eta_j\xi_{j+1,j}}\right), \tag{68}$$

contributes. There are two saddle-points with $\text{sgn}(\phi_{j+1,j})$ for $1 \leq j \leq n-1$ either all positive

or negative, and otherwise specified by

$$s_* = s_1^* = \cdots = s_{n-1}^* = \frac{2\pi\gamma}{L}, \tag{69}$$

$$t_{n,n-1}^* = \cdots = t_{32}^* = t_{21}, \quad \sqrt{\frac{\xi_{j+1,j}^*}{\eta_*}} = (T_b)_{j+1,j}, \quad \text{for } 1 \leq j \leq n-1. \tag{70}$$

The relations (70) express the linearity of the classical motion of the particle in flat space. After expanding the exponent (68) to quadratic order about its saddle-point and integrating over $t_{n,n-1}, \ldots, t_{32}, s$, and $s_1, \ldots, s_{n-1}$, we indeed obtain that the $n$-point probability distribution factorizes as

$$\int q_{T_{n,n-1}, \cdots, T_{21}}(dx_n, \ldots, dx_1) \approx \frac{1}{2} \int_{\substack{(x_1,x_2),\ldots,(x_{n-1},x_n)\in\text{region 4} \\ \cup (x_1,x_2),\ldots,(x_{n-1},x_n)\in\text{region 3}}} \gamma^{-1} d\xi_{n,n-1} \cdots \gamma^{-1} d\xi_{21} \prod_{j=1}^{n-1} f_{T_{j+1,j}}(\xi_{j+1,j}). \tag{71}$$

## 4.3 Evolution of probability

Having verified that the classical limit of the quantum stochastic process given by (47), (48) is Markovian, we now turn to considering an equation involving the quantum analogue of the generator for a Markovian process discussed near (38). Such an equation would express the instantaneous evolution of probability under the action of the quantum kernel (52). We will again work with the flat limit of the particle propagator given in (57) which does not include corrections due to the curvature of $\widetilde{AdS}_2$.

For a quantum stochastic process with joint quantum distributions $q_{T_n,\ldots,T_1}(dx_n, \ldots, dx_1)$, the action of the generator of the process on some probability measure $v(dx)$ would be given by (cf. (38))

$$(QG \cdot v)(dx_3) = \lim_{T_{21}\to 0^+} \partial_{T_2} \int_{x_1} \frac{q_{T_2,T_1}(dx_3, dx_1)}{q_{T_1}(dx_1)} v(dx_1). \tag{72}$$

Furthermore, barring some intrinsic time evolution of the probability measure $v$, we may equate the above action of the generator with that of the direct action of the time-derivative of the quantum kernel $\kappa_{T_2,T_1}(dx_2; x_1) = q_{T_2,T_1}(dx_2, dx_1)/q_{T_1}(dx_1)$ on $v$,[26]

$$\lim_{T_{21}\to 0^+} \int_{x_1} \frac{\partial_{T_2} q_{T_2,T_1}(dx_3, dx_1)}{q_{T_1}(dx_1)} v(dx_1) = \lim_{T_{21}\to 0^+} \partial_{T_2} \int_{x_1} \frac{q_{T_2,T_1}(dx_3, dx_1)}{q_{T_1}(dx_1)} v(dx_1). \tag{73}$$

In the quantum theory of the boundary of JT gravity, the non-fluctuating sub-measure (46) factors out of joint quantum distributions as well as any probability measure — in particular, we may write $v(dx) = \mathcal{D}x\, \Phi(x)$ for some probability *density* $\Phi(x)$. Furthermore, the fact that $q_{T_1}(x_1)$ is infinite and should be factored out consistently between both sides of (73), as a coordinate's worth of volume between points at times $T_1$ and $T_2$, compels us to put the derivative on the right-hand-side in calculable form as

$$\partial_{T_2} \int \frac{q_{T_2,T_1}(dx_3, dx_1)}{q_{T_1}(dx_1)} v(dx_1) = \lim_{T_{32}\to 0^+} \frac{1}{T_{32}} \int \left( \frac{q_{T_3,T_1}(dx_3, dx_1)}{q_{T_1}(dx_1)} - \frac{q_{T_2,T_1}(dx_3, dx_1)}{q_{T_1}(dx_1)} \right) v(dx_1)$$

$$= \lim_{T_{32}\to 0^+} \frac{1}{T_{32}} \left( \iint \frac{q_{T_3,T_2,T_1}(dx_3, dx_2, dx_1)}{q_{T_1}(dx_1)} v(dx_1) - \int \frac{q_{T_2,T_1}(dx_3, dx_1)}{q_{T_1}(dx_1)} v(dx_1) \right). \tag{74}$$

---

[26]The two actions are a priori distinct as integration and differentiation do not commute when the integrand is singular in the relevant domain, and $\lim_{T_2\to T_1} \kappa_{T_2,T_1}(dx_2; x_1)$ is singular, e.g. see below (52).

The resulting *generator equation*, expressing a constraint on probability measures which evolve in time according to joint quantum distributions of the boundary of JT gravity, is given by (5), where we have also conveniently Taylor-expanded $\Phi(x_1)$ in the first integral of the right-hand-side.

We now proceed to evaluate (5) for the *asymptotic* quantum stochastic process we extracted from the *local* limit of the quantum theory of the boundary of JT gravity in Section 4.2.2, working in the holographic, semi-classical limit $\gamma \gg s \gg 1$ and expanding in large $\gamma$. We will find that in the first non-vanishing order which is at $\gamma^0$, we recover Einstein's equations (2) with the cosmological constant $\Lambda$ set to zero, with the probability density $\Phi$ being the dilaton, or area of compactified space, in the gravitational theory!

In detail, we first evaluate integrals on each side of (5) for finite $T_{21} > 0$ and $T_{32}, T_{21} > 0$, where in the semi-classical limit the propagator (57) applies (and enters the joint quantum distributions $q_{T_3,T_2,T_1}$, $q_{T_2,T_1}$ as in (20), (21), (22), (27)) and we can also use the saddle-point method. We then continue the resulting expressions to $T_{21} \to 0^+$ and $T_{21}, T_{32} \to 0^+$. We need to go up to sub-subleading order in the saddle-point expansion, so it is convenient to use a closed formula for the sadde-point expansion of a multi-variable integral, which we derive in Appendix E. For $\mathcal{N}$ variables $z_1, \ldots, z_{\mathcal{N}}$,

$$
\int d\boldsymbol{z} \left( \sum_{|\boldsymbol{k}|=0}^{\infty} f_{\boldsymbol{k}} (\boldsymbol{z} - \boldsymbol{z}_*)^{\boldsymbol{k}} \right) \exp\left( -\left( p(\boldsymbol{z}_*) + \sum_{\mathcal{M}=1}^{\mathcal{N}} p_{\mathcal{M}} \left( z_{\mathcal{M}} - z_{\mathcal{M}}^* \right)^2 + \sum_{|\boldsymbol{k}|=3}^{\infty} p_{\boldsymbol{k}} (\boldsymbol{z} - \boldsymbol{z}_*)^{\boldsymbol{k}} \right) \right)
$$

$$
= e^{-p_*} \sum_{m=0}^{\infty} \sum_{j=0}^{2m} \sum_{|\boldsymbol{i}|=3j}^{2(m+j)} \frac{(-1)^j}{j!} \hat{B}_{\boldsymbol{i}j}(p) \sum_{|\boldsymbol{k}|=2(m+j)-|\boldsymbol{i}|} f_{\boldsymbol{k}} \prod_{\mathcal{M}=1}^{\mathcal{N}} \frac{1}{2} \left( 1 + (-1)^{(\boldsymbol{k}+\boldsymbol{i})_{\mathcal{M}}} \right) p_{\mathcal{M}}^{-((\boldsymbol{k}+\boldsymbol{i})_{\mathcal{M}}+1)/2} \Gamma\left( \frac{(\boldsymbol{k}+\boldsymbol{i})_{\mathcal{M}}+1}{2} \right),
$$

$$\tag{75}$$

where the star scripts indicate evaluation at the saddle point, and $\hat{B}_{\boldsymbol{i}j}(x)$, $x : \mathbb{N}^{\mathcal{N}} \to \mathbb{C}$ are multi-variate Bell polynomials [22] which are coefficients in the expansion of powers of a multi-variate polynomial—for $S = \sum_{\boldsymbol{i} \in \mathbb{N}^{\mathcal{N}}} x_{\boldsymbol{i}} \boldsymbol{z}^{\boldsymbol{i}}$, $S^j = \sum_{\boldsymbol{i} \in \mathbb{N}^{\mathcal{N}}} \hat{B}_{\boldsymbol{i}j}(x) \boldsymbol{z}^{\boldsymbol{i}}$ for $j \in \mathbb{N}$. The Bell polynomials can be evaluated using the recursion relation

$$
\hat{B}_{\boldsymbol{i}j}(x) = \begin{cases} \delta_{\boldsymbol{i},\boldsymbol{0}}, & j = 0, \\ x_{\boldsymbol{i}}, & j = 1, \\ \displaystyle\sum_{|\boldsymbol{r}|=J(j-1)}^{|\boldsymbol{i}|-J} \hat{B}_{\boldsymbol{i},j-1}(x) \hat{B}_{\boldsymbol{i}-\boldsymbol{r},1}(x), & j \geq 2, \end{cases} \tag{76}
$$

where $J$ is an integer s.t. $x_{\boldsymbol{i}} = 0$ for $|\boldsymbol{i}| < J$. In our case with $x_{\boldsymbol{i}} = p_{\boldsymbol{i}}$, $J = 3$. Let us also note the formula (75) assumes that the matrix of second derivatives of the exponent $p$ is diagonal in the variables $\boldsymbol{z}$.

We can then evaluate (5) in the semi-classical limit to $O(\gamma^0)$ accuracy, via calculations similar to those outlined in (60), (62) and (67), (68). See Appendix F for details. Some points of note are i) saddle-points exist, and thus the integrals are supported, only in the relative regions $(x_2; x_1) \in$ region 3 (or 4), and $(x_3; x_2), (x_2; x_1) \in$ region 3 (or 4) (see Figure 7), ii) we should make the exponents which appear in the integrals analytic, by changing variables from $\xi_{j,i}$ to the scaled geodesic distance $l_{j,i} = \sqrt{\xi_{j,i}}$, iii) in applying (75) variables must be used which diagonalize the matrix of second derivatives of the exponent, iv) the infinite factor $q_{T_1}(x_1) \propto \text{vol}(H(x_1))$ must be factored out consistently across terms in (5), and v) in evaluating the first integral on the RHS of (5), we convert the Taylor expansion of $\Phi$ in absolute coordinates to an expansion in relative coordinates, using the flat geometry at short distances (58).

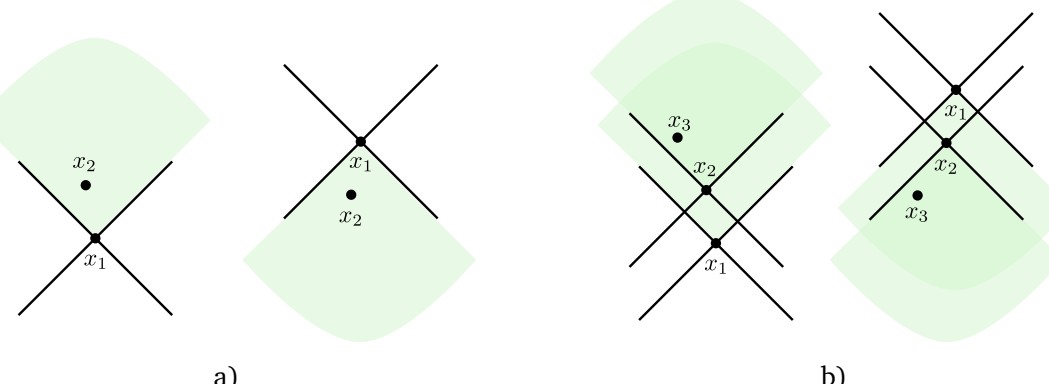

Figure 7: Relative regions contributing in (5) to a) the integral on the LHS and the second integral on the RHS, and b) the first integral on the RHS, with $T_{32}$ and $T_{21}$ held finite and positive.

Collecting terms to one side, we obtain

$$\sum_{(x;x')\in\,\text{regions}\,3,4} \lim_{x'\to x} \frac{i\gamma}{8} l^{-2}\partial_t^2 \Phi(l,t) = 0, \tag{77}$$

where $l, t$ are relative coordinates of $(x; x')$ given by (25), (26), (56), and

$$l = \sqrt{\xi}. \tag{78}$$

As a final step, let us write the equation (77) with respect to absolute coordinates $X^\mu$ of $x$. It is convenient to employ the null coordinates $x = (\varphi_1, \varphi_2), x' = (\varphi_3, \varphi_4)$ defined in (25). Then in our limit of interest $x' \to x$, we find

$$\partial_l \to \pm\sqrt{\frac{2}{\gamma}}\cos\theta(e^{-t}\partial_{\varphi_1} + e^t\partial_{\varphi_2}), \qquad \begin{pmatrix} \text{upper sign:}\ (x;x') \in \text{region 3} \\ \text{lower sign:}\ (x;x') \in \text{region 4} \end{pmatrix}, \tag{79}$$

cancel between regions 3 and 4, and[27]

$$l^{-2}(\partial_t^2 - l\partial_l) \to \frac{\partial X^\mu}{\partial l}\frac{\partial X^\nu}{\partial l}(\nabla_\mu\nabla_\nu - g_{\mu\nu}\nabla^2), \tag{80}$$

where the tensorial components $(\partial X^\mu/\partial l)(\partial X^\nu/\partial l)$ vary independently as we vary $t$, for example $\partial\varphi_1/\partial l = \pm\sqrt{\frac{2}{\gamma}}\cos\theta e^{-t}$, $\partial\varphi_2/\partial l = \pm\sqrt{\frac{2}{\gamma}}\cos\theta e^t$. The relative coordinate $t$ corresponds to the "direction of inflow of probability" we are examining (see Figure 8), and was fixed to an arbitrary value by factoring out the volume of the symmetry group $\text{vol}(H(x_1))$ in (5). Thus we conclude that for the generator equation

$$\sum_{(x;x')\in\,\text{regions}\,3,4} \lim_{x'\to x} \frac{i\gamma}{8} l^{-2}\partial_t^2 \Phi(l,t) = \frac{i\gamma}{4}\frac{\partial X^\mu}{\partial l}\frac{\partial X^\nu}{\partial l}(\nabla_\mu\nabla_\nu - g_{\mu\nu}\nabla^2)\Phi(x) = 0, \tag{81}$$

to hold independently of the arbitrary choice, its individual components must vanish. The latter are Einstein's equations of JT gravity with zero cosmological constant, once we identify the constrained probability density $\Phi$ of the quantum theory with the area of compactified space in the gravitational theory.

---

[27]In the following, we are able to retain Christoffel symbols in covariant derivatives corresponding to the curvature of AdS$_2$, by taking the asymptotic limit $x' \to x$ *after* taking derivatives involved in rewriting $\partial_t^2$, $\partial_t^2 = \frac{\partial X^\mu}{\partial t}\frac{\partial X^\nu}{\partial t}\partial_\mu\partial_\nu + \frac{\partial X^\mu}{\partial t}\partial_\mu\left(\frac{\partial X^\nu}{\partial t}\right)\partial_\nu.$

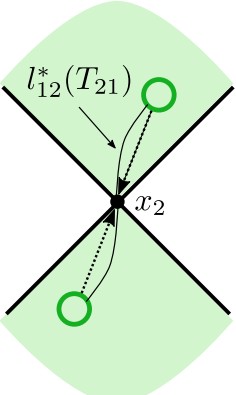

Figure 8: Depiction of the evolution of probability under the quantum stochastic process, $\left(QM_{T_2,T_1}\Phi\right)(x_2) \equiv \int \mathcal{D}x_1 \frac{q_{T_2,T_1}(x_2,x_1)}{q_{T_1}(x_1)}\Phi(x_1)$, in the semi-classical limit. The original probability is displaced by the average of probabilities at distance $l_{21}^*(T_{21})$ determined by the saddle point of the integral, with the direction of inflow fixed by our choice of $t_{21} = t$.

Let us clarify and further discuss several aspects of our results so far. First, it a non-trivial check of the robustness of our construction that we can reproduce the Christoffel symbols inside covariant derivatives in (81), when using the full relative coordinate $t$ in $\widetilde{AdS_2}$ defined in (26). See footnote 27. However, to be consistent with our use of the asymptotic, flat propagator (57) we should only use the flat asymptotics of $l, t$ given in (58), i.e. take the limit $x' \to x$ before rewriting the derivative $\partial_t^2$ using absolute coordinates. Then what we recover in (81) are in fact Einstein's equations in flat JT gravity.

Let us recap the logic of our deconstruction which has led us to identify Einstein's equations in flat JT gravity as the leading semi-classical approximation to an exact quantum equation. We started out with a quantum stochastic process defined by EVPP's (47), (48) governing the position observable in the quantum theory of the boundary of anti-de Sitter JT gravity. We then extracted the flat asymptotics of the quantum propagator of the observable, and therefore of the EVPP's. This gave us a consistent set of joint quantum distributions defining a new quantum stochastic process, of which the generator equation gives Einstein's equations in flat JT gravity, with a dilaton configuration of the latter being identifiable as probability density evolving under the quantum stochastic process.

Our framework of quantum stochastic processes is thus seen to give a natural answer to what has been a long-term puzzle: How the main implication of AdS/CFT, namely that gravity arises from quantum physics without gravity, can be extended to gravity in flat space where there is no time-like boundary where a "dual" quantum system can reside.[28] Our answer is that gravity fundamentally has to do with quantum stochastic processes, where the latter can be induced by a quantum system—i.e. the joint quantum distributions computed using the Hilbert space, density matrix, Hamiltonian, and projectors of an actual quantum system as in (4)—but that a set of consistent joint quantum distributions, which in particular satisfy marginalization relations, can be specified without referring to any such quantum system.[29] The time evolution of joint quantum distributions results in a generator equation for the process, which we are proposing is the quantum completion of Einstein's equations.

---

[28]In the case of flat JT gravity, after setting the cosmological constant $\Lambda$ to zero in (1), the rigid two-dimensional space is flat and the boundary action involving curvature disappears.

[29]Recall from below (43) that other consistency conditions we have imposed are that each distribution $q_{T_n,...,T_1}(x_n,...,x_1)$ integrates to one, and that the distribution at a single time $q_{T_1}(x_1)$ is positive.

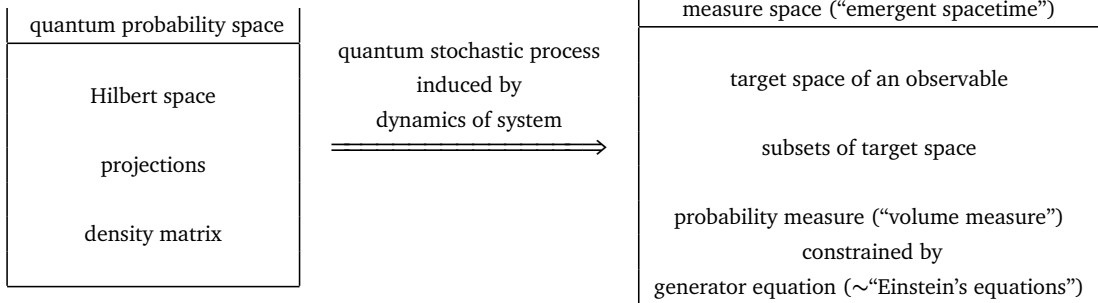

Figure 9: A recasting of holographic duality between a quantum system and a gravitational theory.

Let us also briefly comment on the form of quantum corrections to Einstein's equations in JT gravity, i.e. the generator equation (5) at higher orders in $\gamma^{-1}$. At $O(\gamma^0)$ at which we recovered Einstein's equations, we see only up to second derivatives of $\Phi$. At higher orders, higher-order derivatives of $\Phi$ will appear, as among other effects, powers of $l_{32}$ (which translate to powers of $T_{32}$) are stripped away by the taking of derivatives in the saddle-point expansion in (75).

Finally, extrapolating our analysis of JT gravity, we conjecture the following. General relativity arises in the semi-classical limit of the evolution of probability with respect to quantum stochastic processes, with the "volume" measure of "spacetime" being a probability measure in the target space of a quantum observable. In contrast to the limiting case of JT gravity we have studied above, in general, the probability measure will not only evolve under but also enter the quantum stochastic process governing the observable, and therefore satisfy a non-linear generator equation. The generator equation expresses the time evolution of joint quantum distributions, and provides a quantum completion of Einstein's equations, which are none other than components of the generator equation in the leading non-vanishing order in the semi-classical limit.

## 5 Conclusion and discussion

Motivated by holographic duality and the Ryu-Takayanagi formula, and by the simple, controlled setting afforded by JT gravity and the quantum description of its boundary degrees of freedom, we have attempted to elucidate how quantum theory can give rise to gravity. We have proposed a framework based on defining quantum stochastic processes, which are analogous in appropriate ways to stochastic processes in classical probability theory, and discovered that it is broad enough to encompass both gravity with and without a cosmological constant.

In this framework, the volume measure of a spacetime is identified as a probability measure in the target space of a quantum observable, and is constrained with respect to the quantum stochastic process governing the observable, with the generator equation of the process providing a quantum completion of Einstein's equations. As a by-product of our proposal, we are able to demystify or recast holographic duality as a mapping from a quantum probability space to a measure space, see Figure 9.

One question that may arise is the role in our discussion of the microscopic theory, the SYK model, for which JT gravity is a low-energy effective theory. It appears that the information about the background spacetime AdS$_2$ already comes along with the JT gravity action in (1), so what does it mean for the bulk spacetime to emerge? In fact, the background spacetime can be said to emerge from the low-energy dynamics of the microscopic SYK model, with there being

a direct relationship between coordinates of the bulk spacetime and the form of the fermionic Green's function in SYK. Working in Euclidean signature for simplicity, the low-energy modes of the Green's function $G(\tau_1, \tau_2) = \langle \chi_j(\tau_1) \chi_j(\tau_2) \rangle$ are reparametrizations

$$G(\tau_1, \tau_2) = \widetilde{G}_c(\varphi(\tau_1), \varphi(\tau_2)) \left( J^{-1} \varphi'(\tau_1) \right)^\Delta \left( J^{-1} \varphi'(\tau_2) \right)^\Delta, \tag{82}$$

where $\varphi(\tau)$ assumes values on a circle of length $2\pi$, and $\widetilde{G}_c(\varphi_1, \varphi_2) = b^\Delta \left| 2 \sin \frac{\varphi_1 - \varphi_2}{2} \right|^{-2\Delta}$ is the conformal two-point function [9]. The soft mode $\varphi(\tau)$ is governed by the effective Schwarzian action

$$I_{\text{Sch}}[\varphi] = -\gamma \int_0^L \text{Sch}(e^{i\varphi}, \ell) \, d\ell, \tag{83}$$

with $\gamma$ and $L$ being related to parameters $N$ and $\beta$ in the SYK model as in (18). Now, we can identify $\varphi(\tau)$, $\varphi'(\tau)$ as bulk coordinates near the boundary of the 2-dimensional hyperbolic disk, where fixing the metric as

$$ds^2 = \frac{4(dr^2 + r^2 d\varphi^2)}{(1 - r^2)^2}. \tag{84}$$

$\varphi$ is the angular coordinate and $\varphi'$ is related to the radial coordinate $r$ as $\varphi = \gamma(1 - r)$. That is, if we had a free field $\chi$ in the bulk with $sl_2$ Casimir eigenvalue $\lambda$, its two-point function $\langle \chi(X_1) \chi(X_2) \rangle$ at coordinates $X_1 = (\varphi(\tau_1), \varphi'(\tau_1))$, $X_2 = (\varphi(\tau_2), \varphi'(\tau_2))$ would be given by the RHS of (82).

Now, what is the relationship between the effective Schwarzian action (83) and the JT gravity action (1)? The Schwarzian action can be derived from the SYK model when one works in the holographic limit $N \gg \beta_{\text{SYK}} J \gg 1$ and simultaneously, at long time scales $|\tau_1 - \tau_2| J \gg 1$ [9]. The JT gravity action can be viewed as a completion of the Schwarzian action to short time scales $|\tau_1 - \tau_2| J \ll 1$. That is, the JT gravity action, with Dirichlet boundary conditions and negative cosmological constant, reduces to the Schwarzian action at long time scales [4]. Equivalently, at the quantum level and in Lorentzian signature, the boundary particle resulting from quantizing the JT action lives in exact AdS$_2$, whereas in the Schwarzian limit the particle lives in an asymptotic geometry near the boundary of AdS$_2$.

Given the above, the new proposal we have made here regarding the "emergence" of spacetime, is to understand the spacetime *operationally* in relation to the quantum-mechanical degrees of freedom resulting from quantizing gravity.[30] In the two-dimensional setting we have analyzed, this means interpreting AdS$_2$ as the target space of the quantum observable resulting from quantizing the boundary action of JT gravity. We attempted to show the utility of this viewpoint by deriving the gravitational equations of motion solving for the dilaton in JT gravity, which has a priori no relationship with the boundary action of JT gravity. Despite there being no such relation, we were able to construct the gravitational equations of motion from scratch, by assuming the effective volume measure at a point (given by the product of the volume measure on AdS$_2$ and an appropriately defined dilaton field) is a probability measure constrained by the quantum degrees of freedom of the boundary.

We have also made the conjecture that this identification of spacetime as the target space of quantum degrees freedom in gravity, and the volume measure as a probability measure constrained by the dynamics of those degrees of freedom, persists beyond JT gravity and in particular to higher dimensions. However, the technical derivation of our results in JT gravity

---

[30]This is a statement about gravity proper and its quantum degrees of freedom, as opposed to any relation between gravity as an effective theory and an underlying microscopic theory. Relations of the latter kind, e.g. those between the Green's function in SYK and bulk coordinates described above, are expected to be model-specific and non-universal. Of course a caveat is that there may be, and in fact are known to be, certain universal ways in which e.g. correlators in the UV couple to effective degrees of freedom of gravity in the IR.

certainly did depend on the details and solvable nature of the theory. In ongoing and future work, we hope to give proposals and theoretical machinery for calculating the relevant physical quantities we have identified in this paper, for example joint quantum distributions, in higher-dimensional gravity.

Let us also remark on the relationship between our work and previous work on deriving Einstein equations from entanglement entropy, see [23]. In technical terms, the work in [23] crucially relied on an expansion around the CFT vacuum and had the limitation of obtaining only the linearized Einstein's equations. Conceptually, entanglement entropy only characterizes the quantum state, and it is natural to expect that the dynamics of the boundary quantum system should be accounted for in order to fully reproduce Einstein's equations. In our work we are proposing to quantify the necessary quantum dynamics using joint quantum distributions as in (4).

Finally, we outline three further lines of inquiry to be pursued in the future.

- Significance of Markovianity in the classical limit

  In the setting of flat JT gravity, we have discovered that the stochastic process of the theory is characterized by Markovianity in the classical limit. In [5] we will characterize to what extent Markovianity in the classical limit holds in curved JT gravity; we suspect it will fail at finite times, only holding in a sense local in time. In general, it will be interesting to see whether some degree of Markovianity could be a precondition of the probability residing in a measurable space to form a manifold corresponding to "spacetime".

- Probabilistic interpretation of action principle for gravity

  In our deconstruction, we have reproduced Einstein's equations from the generator equation of a quantum stochastic process. However, traditionally, general relativity has been formulated with an action, which one varies with respect to the metric to obtain Einstein's equations. Thus it is natural ask to what the interpretation of this action principle is in our probabilistic framework. More concretely, we can ask about the interpretation of the curvature scalar, and the metric.

  With this information, one may attempt—in a general theory of gravity—to either reconstruct the generator equation, or build an effective quantum theory order by order in the semi-classical expansion.

  Relatedly, one would like a probabilistic interpretation of the energy-momentum tensor, which may be explored by turning on sources in the Hamiltonian of the boundary theory of anti-de Sitter JT gravity, and tracing its effects to the generator equation.

- Derivation of the Ryu-Takayanagi formula

  Previously, the Ryu-Takayanagi formula and its generalizations have been argued for (see e.g. [24]) by assuming the existence of a duality between a gravitational theory and a quantum system on its boundary. It will be interesting to derive it from our direct identification (in anti-de Sitter gravity) of the volume measure as a probability measure satisfying a generator equation determined by, among other ingredients, the density matrix of a corresponding quantum system. (Integration over a codimension-two surface produces a probability density in the quantum theory.)

# Acknowledgments

We thank Lars Bildsten, David Gross, and Ian Moult for support and encouragement, and Sergei Dubovsky, David Gross, and Erik Verlinde for valuable comments.

**Funding information** This work was supported by a grant to the KITP from the Simons Foundation (#216179), by the National Science Foundation under Grant No. NSF PHY-1748958, and by the Simons Foundation through the "It from Qubit" Simons collaboration.

# A Quantum probability space and observables

We summarize definitions of a quantum probability space and of a quantum observable which are formulated [20] in analogy to notions of a probability space and random variable in classical probability theory. See Section 3 for a summary of the latter.

## A.1 Quantum probability space $(\mathcal{H}, \mathcal{P}(\mathcal{H}), \rho)$

- $\mathcal{H}$: Hilbert space

  An operator on $\mathcal{H}$ is a linear map on $\mathcal{H}$. The norm of an operator $T$ is given by

  $$\|T\| = \sup_{\|u\|=1} \|Tu\|. \tag{A.1}$$

  A sequence $\{T_n\}$ is said to converge in operator norm if

  $$\lim_{n\to\infty} \|T_n - T\| = 0. \tag{A.2}$$

  It is said to converge strongly if

  $$\lim_{n\to\infty} \|T_n u - Tu\| = 0, \quad \forall u \in \mathcal{H}, \tag{A.3}$$

  and we write $\text{s.lim}_{n\to\infty} T_n = T$. It is said to converge weakly if

  $$\lim_{n\to\infty} \langle u, T_n v \rangle = \langle u, Tv \rangle, \quad \forall u, v \in \mathcal{H}, \tag{A.4}$$

  and we write $\text{w.lim}_{n\to\infty} T_n = T$.

- $\mathcal{P}(\mathcal{H})$: Set of all projections on $\mathcal{H}$

  An operator $T$ on $\mathcal{H}$ is a projection iff

  $$T = T^* = T^2. \tag{A.5}$$

  Note $T$ is a positive, self-adjoint, bounded operator.

  A projection $E$ corresponds to an event. Let us denote range of $E$ by $R(E)$. Given a family of projections $E_\alpha$,

  $$\bigvee_\alpha E_\alpha : \text{Projection on the smallest closed subspace containing } \bigcup_\alpha R(E_\alpha)$$
  $$\bigwedge_\alpha E_\alpha : \text{Projection on the smallest closed subspace containing } \bigcap_\alpha R(E_\alpha) \tag{A.6}$$

  are operations analogous to taking the intersection and union of events as sets in classical probability. Simple limiting cases are

– A sequence of events $\{E_n\}$ have ranges which are mutually orthogonal, i.e.

$$E_i E_j = 0, \quad \forall i \neq j \quad \Rightarrow \quad \bigvee_n E_n = E_1 + E_2 + \cdots$$

– $\{E_n\}$ are mutually commuting $\quad \Rightarrow \quad \bigwedge_n E_n = \mathrm{s.lim}_n E_1 E_2 \ldots E_n.$

- $\rho$: A positive operator with unit trace (density matrix)

A probability distribution on $\mathcal{P}(\mathcal{H})$ is a map

$$\mu : \mathcal{P}(\mathcal{H}) \to [0,1], \tag{A.7}$$

s.t. for every sequence $\{\mathcal{P}_n\}$ in $\mathcal{P}(\mathcal{H})$ which have ranges which are mutually orthogonal,

$$\mu\left(\bigvee_n P_n\right) = \mu\left(\sum_n P_n\right) = \sum \mu(P_n), \tag{A.8}$$

and $\mu(\mathbf{1}) = 1$. A probability distribution on $\mathcal{P}(\mathcal{H})$ with $\dim \mathcal{H} \geq 3$ is in one-to-one correspondence with a density matrix,

$$\mu(P) = \mathrm{tr}(\rho P), \quad \forall P \in \mathcal{P}(\mathcal{H}). \tag{A.9}$$

## A.2 Observables

- Given a measurable space $(\Omega, \mathcal{F})$, an $\Omega$-valued observable $\xi$ is a projection-valued measure on $(\Omega, \mathcal{F})$, i.e. a mapping

$$\xi : \mathcal{F} \to \mathcal{P}(\mathcal{H}), \tag{A.10}$$

satisfying $\xi(\bigcup_j F_j) = \sum_j \xi(F_j)$ if $F_i \bigcap F_j = \varnothing$ for $i \neq j$, and $\xi(\Omega) = \mathbf{1}$. Note whenever $\Omega$ is a topological space, $\mathcal{F} = \mathcal{F}_\Omega$ can be taken to be the Borel $\sigma$-algebra generated by the open subsets of its topology.

- A bounded normal operator $T$ is in one-to-one correspondence with a bounded $\mathbb{C}$-valued observable $\xi : \mathcal{F}_{\mathbb{C}} \to \mathcal{P}(\mathcal{H})$ via

$$T = \int_{\{z \mid |z| \leq \|T\|\}} z\, \xi(dz), \tag{A.11}$$

where

$$\begin{aligned}
&T \text{ unitary} \Longleftrightarrow \xi \text{ is supported on } \{z : |z| = 1\}, \\
&T \text{ self-adjoint} \Longleftrightarrow \xi \text{ is supported on } [-\|T\|, \|T\|], \\
&T \text{ positive} \Longleftrightarrow \xi \text{ is supported on } [0, \|T\|].
\end{aligned} \tag{A.12}$$

A self-adjoint operator $T$ (not necessarily bounded) is in one-to-one correspondence with a $\mathbb{R}$-valued observable $\xi : \mathcal{F}_{\mathbb{R}} \to \mathcal{P}(\mathcal{H})$ via

$$T = \int_{\mathbb{R}} x\, \xi(dx). \tag{A.13}$$

- Canonical $\Omega$-valued observable

Let $(\Omega, \mathcal{F}, \mu)$ be a $\sigma$-finite measure space where $\mathcal{F}$ is countably generated. Note on a $\sigma$-finite measure space the Hilbert space $L^2(\mu)$ is the space of ($\mu$-equivalent classes of) complex-valued, absolutely square-summable functions on $\Omega$ with inner product

$$\langle f, g \rangle = \int f^*(x) g(x) \mu(dx), \quad f, g, \in L^2(\mu). \tag{A.14}$$

The canonical $\Omega$-valued observable $\xi^{\mu} : \mathcal{F} \to \mathcal{P}(\mathcal{H})$ is given by

$$(\xi^{\mu}(B)f)(w) = I_B(w)f(w), \quad f \in L^2(\mu), \tag{A.15}$$

where $I$ is the indicator function. For example, let $\Omega$ be $\widetilde{AdS}_2$, with $\mu(d^2x) = d^2x\sqrt{-g}$. Then

$$(\xi^{\mu}(B)f)(y) = \int_B \delta(x-y)f(y) = \begin{cases} f(y), & \text{if } y \in B, \\ 0, & \text{if } y \notin B. \end{cases} \tag{A.16}$$

Writing $f(x) = \langle x | f \rangle$, from (A.14) $\mathbf{1} = \int \mu(d^2x)|x\rangle\langle x|$ and

$$\int \mu(d^2x)|x\rangle\langle x|y\rangle = |y\rangle \Rightarrow \langle x|y\rangle = \delta^2(x-y)/\sqrt{-g},$$

so that the canonical variable $\xi^{\mu}$ is given by

$$\xi^{\mu}(d^2x) = \mu(d^2x)|x\rangle\langle x|. \tag{A.17}$$

# B Propagator of boundary particle in local limit

## B.1 Asymptotic propagator at short distances

For $(x;x') \in$ regions 3, 4, the two-point function of the boundary particle in JT gravity is given in terms of $\check{C}_{\lambda,\nu}(w)$ in (30). We will obtain an asymptotic form for $\check{C}_{\lambda,\nu}(w)$ by taking the limit $\gamma^2 \gg s^2 \gg 1, z \ll 1$ in the differential equation solved by the function.

The regularized hypegeometric function $\mathbf{F}(a,b,c;z)$ solves the differential equation

$$\left( \partial_z^2 + \frac{(c-(a+b+1)z)}{z(1-z)}\partial_z - \frac{ab}{z(1-z)} \right)\mathbf{F}(z) = 0. \tag{B.1}$$

To eliminate the first-order derivative term, we define the function

$$h(z) = \mathbf{F}(z)(1-z)^{-\alpha}z^{-\beta}, \qquad \alpha = \frac{c-(a+b+1)}{2}, \quad \beta = -\frac{c}{2}, \tag{B.2}$$

which solves the equation

$$h''(z) = \left( \frac{\alpha(\alpha+1)}{(1-z)^2} + \frac{(ab+\alpha c)}{z(1-z)} - \frac{(\beta^2+(c-1)\beta)}{z^2} \right)h(z). \tag{B.3}$$

For the hypergeometric function appearing in

$$\check{A}_{\lambda,\nu,-\nu}(w) = (1-z)^{-\nu}\mathbf{F}(\lambda-\nu,1-\lambda-\nu,1;z), \tag{B.4}$$

with $a = \lambda - \nu$, $b = 1 - \lambda - \nu$, $c = 1$ and $\alpha = \nu - \frac{1}{2}$, $\beta = -\frac{1}{2}$, we have

$$h''(z) = \left( \gamma^2 f(z) + g(z) \right)h(z),$$
$$f(z) = -\frac{1}{z(1-z)^2}, \quad g(z) = \frac{\lambda(1-\lambda)}{z(1-z)} - \frac{1}{4z^2(1-z)^2}. \tag{B.5}$$

Next, we transform the differential equation s.t. the large parameter $\gamma$ multiplies a simple pole plus a constant: Setting $H = \left( \frac{dz}{d\zeta} \right)^{-1/2} h$,

$$\frac{d^2H}{d\zeta^2} = \left( \gamma^2 f(z)\dot{z}^2 + \psi(\zeta) \right)H,$$
$$\psi(\zeta) = g(z)\dot{z}^2 + \dot{z}^{1/2}\frac{d^2}{d\zeta^2}\left( \dot{z}^{-1/2} \right), \tag{B.6}$$

where $\zeta$ is determined by $f(z)\dot{z}^2 = -\left(\frac{1}{\zeta} + c\right)$ and $c$ is a constant we can choose. Then

$$z = \zeta + \frac{1}{3}(-2 + c)\zeta^2 + \frac{1}{45}(17 - c(20 + c))\zeta^3 + O(\zeta^4), \tag{B.7}$$

and

$$\gamma^2 f(z)\dot{z}^2 + \psi(\zeta) = -\frac{1}{4\zeta^2} - \frac{\left(\gamma^2 - \lambda(1 - \lambda) + \frac{c+1}{6}\right)}{\zeta} - c\left(\gamma^2 - \lambda(1 - \lambda) + \frac{1}{6}\right)$$
$$- \lambda(1 - \lambda) + \frac{9c^2}{20} + \frac{11}{60} + O(\zeta). \tag{B.8}$$

We choose $c = 1$ so that the $O(\zeta^0)$ term in (B.8) does not depend on $\lambda$. (This ensures that the scaling of the argument of the two-point function does not depend on the energy of the particle.) We also assume $s^2 \gg 1$ and $\zeta \ll 1$, and choose to neglect $O(s^2\zeta)$ terms. This gives

$$\frac{d^2 H}{d\zeta^2} \approx \left(-\frac{1}{4\zeta^2} - \frac{(\gamma^2 - s^2)}{\zeta^2} - \gamma^2\right)H. \tag{B.9}$$

The equation (B.9) has two independent solutions which are Whittaker functions,

$$M_{-i\eta,0}(i\xi), \quad W_{-i\eta,0}(i\xi), \tag{B.10}$$

where

$$\eta = \frac{\gamma^2 - s^2}{2\gamma}, \qquad \xi = 2\gamma\zeta. \tag{B.11}$$

We can match their small-$z$ asymptotics with that of $\check{C}_{\lambda,\nu}(w)$:

$$W_{-i\eta,0}(i\xi) \sim \frac{(-i\xi)^{1/2}}{\Gamma\left(\frac{1}{2} + i\eta\right)}\left(\ln(i\xi) + \psi\left(\frac{1}{2} + i\eta\right) - 2\psi(1)\right) + \cdots,$$
$$M_{-i\eta,0}(i\xi) \sim (i\xi)^{1/2} + \cdots, \tag{B.12}$$

while

$$\check{C}_{\lambda,\nu}(w) \approx \ln z - 2\psi(1) + \frac{\psi(\lambda + \nu) + \psi(1 - \lambda + \nu) + \psi(\lambda - \nu) + \psi(1 - \lambda - \nu)}{2}. \tag{B.13}$$

Then noting $\frac{1}{2}(\psi(\lambda + \nu) + \psi(1 - \lambda + \nu) + \psi(\lambda - \nu) + \psi(1 - \lambda - \nu)) = \ln(2\gamma\eta) + O\left(\gamma^{-2}\right)$, we have to leading order in small $z$,

$$\check{C}_{\lambda,\nu}(w) \approx \frac{\Gamma\left(\frac{1}{2} + i\eta\right)}{(-i\xi)^{1/2}}W_{-i\eta,0}(i\xi) + \frac{1}{(i\xi)^{1/2}}M_{-i\eta,0}(i\xi)(-i\pi) + O\left(\gamma^{-2}\right). \tag{B.14}$$

## B.2 Exponential form

To derive a further simplified exponential form of the propagator (B.14), we use integral representations for confluent hypergeometric functions related to Whittaker functions as

$$M_{\kappa,\mu}(z) = e^{-\frac{1}{2}z}z^{\frac{1}{2} + \mu}M\left(\frac{1}{2} + \mu - \kappa, 1 + 2\mu; z\right),$$
$$W_{\kappa,\mu}(z) = e^{-\frac{1}{2}z}z^{\frac{1}{2} + \mu}U\left(\frac{1}{2} + \mu - \kappa, 1 + 2\mu; z\right), \tag{B.15}$$

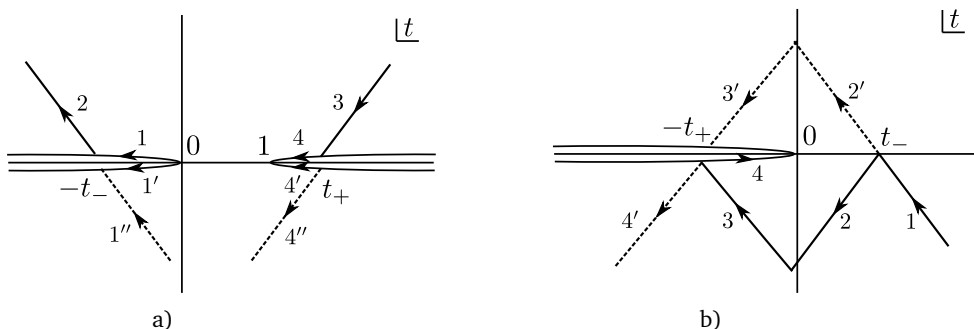

Figure 10: Deformation and substitution of the contour of integration, for a) the integral (B.19), and b) the integral (B.23).

$$M(a, b, z) = \frac{\Gamma(b)}{\Gamma(a)\Gamma(b-a)} \int_0^1 dt\, e^{zt} t^{a-1} (1-t)^{b-a-1}, \qquad \operatorname{Re} b > \operatorname{Re} a > 0,$$

$$U(a, b, z) = \frac{1}{\Gamma(a)} \int_0^{\infty e^{i\phi}} dt\, e^{-zt} t^{a-1} (1+t)^{b-a-1}, \qquad \operatorname{Re} a > 0, -\pi < \phi < \pi \qquad \text{(B.16)}$$

(See Ch. VI of [25], relative to which we have changed notation as $\Phi \to M$, $\Psi \to U$). In (B.16), the functions $t, 1-t$ in the integrand assume their principal values. In terms of the functions (B.15),

$$\check{C}_{\lambda,\nu}(w) \approx -i\pi e^{-\frac{i}{2}\xi} M\left(\frac{1}{2} + i\eta, 1; i\xi\right) - \Gamma\left(\frac{1}{2} + i\eta\right) e^{-\frac{i}{2}\xi} U\left(\frac{1}{2} + i\eta, 1; i\xi\right) + O\left(\gamma^{-2}\right). \quad \text{(B.17)}$$

Note we have the relations

$$\operatorname{ph}\left(e^{-\frac{i}{2}\xi} M\left(\frac{1}{2} + i\eta, 1; i\xi\right)\right) = 0,$$

$$2\operatorname{Im}\left(-\Gamma\left(\frac{1}{2} + i\eta\right) e^{-\frac{i}{2}\xi} U\left(\frac{1}{2} + i\eta, 1; i\xi\right)\right) = \frac{2\pi}{1 + e^{-2\pi\eta}} e^{-\frac{i}{2}\xi} M\left(\frac{1}{2} + i\eta, 1; i\xi\right), \quad \text{(B.18)}$$

which imply (B.17) is real. We proceed to evaluate the integrals for $M$ and $U$ functions using the saddle-point method, valid due to $\gamma \gg 1$, $\eta \gg 1$.

- **$M$ integral**

  We have

  $$e^{-\frac{i}{2}\xi} M\left(\frac{1}{2} + i\eta, 1; i\xi\right) = \underbrace{\frac{1}{\Gamma\left(\frac{1}{2} + i\eta\right)\Gamma\left(\frac{1}{2} - i\eta\right)}}_{\frac{\cosh \pi\eta}{\pi}} \int_0^1 dt\, \underbrace{e^{-\frac{i}{2}\xi + i\xi t}\, t^{-\frac{1}{2} + i\eta} (1-t)^{-\frac{1}{2} - i\eta}}_{e^{-p_M(t)} t^{-1/2}(1-t)^{-1/2}},$$

  $$\text{(B.19)}$$

  where the exponent is

  $$p_M(t) = i\left(\xi\left(\frac{1}{2} - t\right) - \eta\left(\ln t - \ln(1-t)\right)\right), \qquad \text{(B.20)}$$

  and the saddle-points are at

  $$t_* = \pm\frac{1}{2}\left(\sqrt{1 + \frac{4\eta}{\xi}} \pm 1\right) \equiv \pm t_{\pm}, \qquad t_+ > 1, \, t_- > 0. \qquad \text{(B.21)}$$

After deforming the contour of the integral to pass by the saddle-points as shown in Fig. 10 (over the paths $1, 2, 3, 4$), then expressing the integral over pieces of the path in terms of that over substitute pieces ($1 \to 1' \to 1''$, $4 \to 4' \to 4''$), we obtain

$$
e^{-\frac{i}{2}\xi}M\left(\frac{1}{2}+i\eta,1;i\xi\right) = \frac{1}{\sqrt{2\pi\xi\sqrt{1+\frac{4\eta}{\xi}}}}
$$
$$
\times \left\{ e^{i\left(-\frac{\pi}{4}+\frac{\xi}{2}\sqrt{1+\frac{4\eta}{\xi}}+\eta\ln\left(\frac{\sqrt{1+\frac{4\eta}{\xi}}+1}{\sqrt{1+\frac{4\eta}{\xi}}-1}\right)\right)}\left(1-\frac{i}{24\eta}\frac{\frac{12\eta^2}{\xi^2}+\frac{12\eta}{\xi}+1}{\left(1+\frac{4\eta}{\xi}\right)^{3/2}}+O\left(\frac{1}{\gamma^2}\right)\right)+\text{c.c.} \right\}.
$$

(B.22)

We refer to Appendix E for the formula used to calculate the subleading term in the saddle-point expansion.

- $U$ integral

  We have

$$
-\Gamma\left(\frac{1}{2}+i\eta\right)e^{-\frac{i}{2}\xi}U\left(\frac{1}{2}+i\eta,1;i\xi\right)
$$
$$
= \int_{\infty e^{i\phi}}^{0} dt\, \underbrace{e^{-\frac{i}{2}\xi-i\xi t}\,t^{-\frac{1}{2}+i\eta}(1+t)^{-\frac{1}{2}-i\eta}}_{e^{-p_U(t)}t^{-1/2}(1+t)^{-1/2}}, \qquad -\pi < \phi < \pi,
$$

(B.23)

where the exponent is

$$
p_U(t) = i\left(\xi\left(\frac{1}{2}+t\right)-\eta\left(\ln t - \ln(1+t)\right)\right),
$$

(B.24)

and the saddle-points are at

$$
t_* = \frac{1}{2}\left(-1 \mp \sqrt{1+\frac{4\eta}{\xi}}\right) \equiv \mp t_\pm.
$$

(B.25)

After deforming the contour of the integral to pass by the saddle-points as shown in Fig. 10 (over the paths $1, 2, 3, 4$), then expressing the integral over pieces of the path in terms of that over substitute pieces ($2 \to 2'$, $3 \to 3'$, $4 \to 4'$), we obtain

$$
-\Gamma\left(\frac{1}{2}+i\eta\right)e^{-\frac{i}{2}\xi}U\left(\frac{1}{2}+i\eta,1;i\xi\right) = \sqrt{\frac{2\pi}{\xi\sqrt{1+\frac{4\eta}{\xi}}}}
$$
$$
\times \left\{ e^{i\left(\frac{3}{4}\pi-\frac{\xi}{2}\sqrt{1+\frac{4\eta}{\xi}}-\eta\ln\left(\frac{\sqrt{1+\frac{4\eta}{\xi}}+1}{\sqrt{1+\frac{4\eta}{\xi}}-1}\right)\right)}\left(1+\frac{i}{24\eta}\frac{\frac{12\eta^2}{\xi^2}+\frac{12\eta}{\xi}+1}{\left(1+\frac{4\eta}{\xi}\right)^{3/2}}+O\left(\frac{1}{\gamma^2}\right)\right) \right\}.
$$

(B.26)

Using (B.22) and (B.26), we can write (B.17) as

$$
\check{C}_{\lambda,\nu}(w) = \mathrm{Re}\left(\check{C}_{\lambda,\nu}(w)\right) \approx \frac{1}{2}\sqrt{\frac{2\pi}{\xi\sqrt{1+\frac{4\eta}{\xi}}}}
$$

$$
\times \left\{ e^{i\left(\frac{3}{4}\pi - \frac{\xi}{2}\sqrt{1+\frac{4\eta}{\xi}} - \eta\ln\left(\frac{\sqrt{1+\frac{4\eta}{\xi}}+1}{\sqrt{1+\frac{4\eta}{\xi}}-1}\right)\right)} \left(1 + \frac{i}{24\eta}\frac{\frac{12\eta^2}{\xi^2}+\frac{12\eta}{\xi}+1}{\left(1+\frac{4\eta}{\xi}\right)^{3/2}} + O\left(\frac{1}{\gamma^2}\right)\right) + \text{c.c.}\right\}.
$$

$$(\text{B.27})$$

Only retaining lowest-order terms in small $\xi/\eta \approx 4z$,

$$
\boxed{\check{C}_{\lambda,\nu}(w) \approx \frac{\sqrt{\pi}}{(\eta\xi)^{1/4}}\left\{\frac{1}{2}e^{\frac{3}{4}\pi i}e^{-2i\sqrt{\eta\xi}}\left(1 + \frac{i}{16}\frac{1}{\sqrt{\eta\xi}} + O\left(\frac{1}{\gamma^2}\right)\right) + \text{c.c.}\right\}.}
$$

$$(\text{B.28})$$

## C  Classical trajectories of boundary particle

Here we solve for classical trajectories of the boundary particle of JT gravity. In particular, we calculate the invariant distance $z(T_b)$ that the particle travels in bare proper time $T_b$, defined by

$$
z(x;x') = \frac{\varphi_{13}\varphi_{24}}{\varphi_{12}\varphi_{34}}, \tag{C.1}
$$

where we are using coordinates and notation given in (23), (24), and (25).

We use the ambient space for AdS$_2$

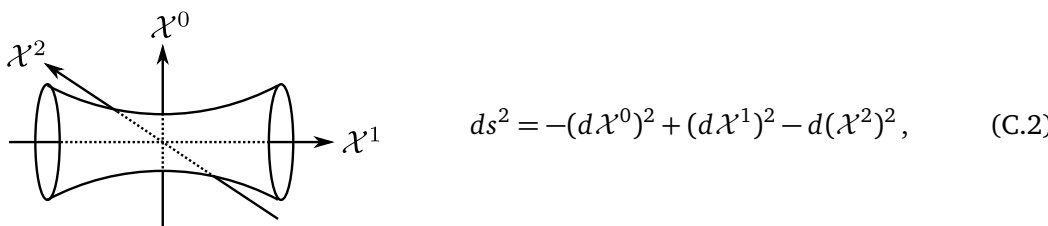

$$
ds^2 = -(d\mathcal{X}^0)^2 + (d\mathcal{X}^1)^2 - d(\mathcal{X}^2)^2, \tag{C.2}
$$

where

$$
\mathcal{X}^0 = \frac{\cos\phi}{\cos\theta}, \quad \mathcal{X}^1 = \tan\theta, \quad \mathcal{X}^2 = \frac{\sin\phi}{\cos\theta}. \tag{C.3}
$$

Vectors in the ambient space and on AdS$_2$ are related by

$$
V^A = e^A_\mu v^\mu, \quad v_\mu = e^A_\mu V_A, \qquad A = 0,1,2, \quad \mu = \phi,\theta, \tag{C.4}
$$

where

$$
e^0_\phi = -\frac{\sin\phi}{\cos\theta} = -\mathcal{X}^2, \qquad\qquad e^0_\theta = \frac{\sin\theta\cos\phi}{\cos^2\theta} = \mathcal{X}^0\mathcal{X}^1,
$$

$$
e^1_\phi = 0, \qquad\qquad e^1_\theta = \frac{1}{\cos^2\theta} = \left(\mathcal{X}^0\right)^2 + \left(\mathcal{X}^2\right)^2,
$$

$$
e^2_\phi = \frac{\cos\phi}{\cos\theta} = \mathcal{X}^0, \qquad\qquad e^2_\theta = \frac{\sin\theta\sin\phi}{\cos^2\theta} = \mathcal{X}^1\mathcal{X}^2. \tag{C.5}
$$

The equations of motion of the boundary particle resulting from the JT action in (1) were obtained in ambient coordinates in Appendix B of [3]. For a time-like trajectory,

$$\begin{cases} \ddot{\mathcal{X}}^A - K\mathcal{N}^A = -\mathcal{X}^A, \\ \mathcal{N}^A - K\mathcal{X}^A = \mathcal{Q}^A = \text{const}, \end{cases} \tag{C.6}$$

where $\dot{\mathcal{X}}^A = e^A_\mu \dot{X}^\mu$, $\mathcal{N}^A = e^A_\mu N^\mu$ are the lifts of unit tangent and normal vectors to the trajectory, and $K$ is its constant curvature. Inserting the second equation into the first, we obtain

$$\boxed{\ddot{\mathcal{X}}^A - (K^2 - 1)\mathcal{X}^A - K\mathcal{Q}^A = 0.} \tag{C.7}$$

The general solution is

$$\boxed{\mathcal{X}^A(T_b) = -\frac{K\mathcal{Q}^A}{K^2 - 1} + c_1^A e^{\sqrt{K^2-1}T_b} + c_2^A e^{-\sqrt{K^2-1}T_b},} \tag{C.8}$$

where in terms of initial conditions $\mathcal{X}^A(0)$, $\dot{\mathcal{X}}^A(0)$,

$$c_1^A = \frac{1}{2}\left(\mathcal{X}^A(0) + \frac{\dot{\mathcal{X}}^A(0)}{\sqrt{K^2-1}} + \frac{K\mathcal{Q}^A}{K^2-1}\right),$$
$$c_2^A = \frac{1}{2}\left(\mathcal{X}^A(0) - \frac{\dot{\mathcal{X}}^A(0)}{\sqrt{K^2-1}} + \frac{K\mathcal{Q}^A}{K^2-1}\right). \tag{C.9}$$

Let us fix initial conditions as follows.

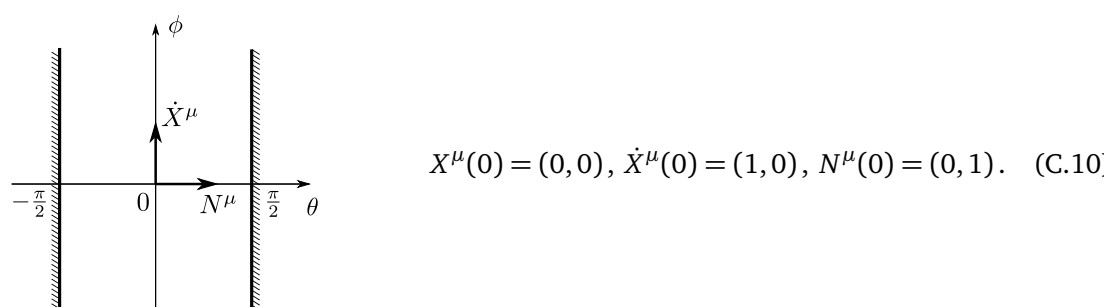

$$X^\mu(0) = (0,0), \ \dot{X}^\mu(0) = (1,0), \ N^\mu(0) = (0,1). \quad (C.10)$$

Then

$$\mathcal{X}^A(0) = (1,0,0), \quad \dot{\mathcal{X}}^A(0) = e^A_\mu \dot{X}^\mu(0) = (0,0,1), \quad \mathcal{N}^A(0) = e^A_\mu N^\mu(0) = (0,1,0),$$
$$\mathcal{Q}^A = \mathcal{N}^A - K\mathcal{X}^A = (-K,1,0). \tag{C.11}$$

Substituting (C.11) into (C.9),

$$c_1^0 = c_2^0 = -\frac{1}{2}\frac{1}{K^2-1}, \quad c_1^1 = c_2^1 = \frac{1}{2}\frac{K}{K^2-1}, \quad c_1^2 = -c_2^2 = \frac{1}{2}\frac{1}{\sqrt{K^2-1}}, \tag{C.12}$$

and

$$\mathcal{X}^0 = \frac{1}{K^2-1}\left(K^2 - \cosh\left(\sqrt{K^2-1}\,T_b\right)\right),$$
$$\mathcal{X}^1 = \frac{K}{K^2-1}\left(-1 + \cosh\left(\sqrt{K^2-1}\,T_b\right)\right),$$
$$\mathcal{X}^2 = \frac{1}{\sqrt{K^2-1}}\sinh\left(\sqrt{K^2-1}\,T_b\right). \tag{C.13}$$

Using (C.10), (C.3), (C.13) in (C.1),

$$z = \frac{1}{2}\left(1 - \frac{\cos\phi}{\cos\theta}\right) = \frac{\sinh^2 \frac{\sqrt{K^2-1}}{2}T_b}{K^2-1}. \tag{C.14}$$

Using the relation between inverse temperature and curvature obtained in [4], $L = \frac{2\pi}{\sqrt{K^2-1}}$, we have

$$\boxed{z(T_b) = \left(\frac{L}{2\pi}\right)^2 \sinh^2\left(\frac{\pi T_b}{L}\right).} \tag{C.15}$$

# D  Proof of Markov property in flat space

Here we show that the Markov property (54) holds for the dynamics of the position observable (boundary) in flat space, using the asymptotic propagator (57), resulting two-event probability distribution (65), and the geometry of flat space (66). Recall when using (57) we also remove the cutoff that is inverse temperature as (64) in the final stage of a calculation.

The saddle-point equations for the integral of the $n$-point EVPP given in (67) are

$$\frac{\partial p}{\partial s} = (\beta + iT_{n,1})s - 2\pi - i\frac{s}{\gamma}\sqrt{\frac{\xi_{n,1}}{\eta}}, \tag{D.1}$$

$$\frac{\partial p}{\partial s_j} = -is_j\left(T_{j+1,j} - \frac{1}{\gamma}\sqrt{\frac{\xi_{j+1,j}}{\eta_j}}\right), \qquad \text{for } i \le j \le n-1, \tag{D.2}$$

$$\frac{\partial p}{\partial \xi_{j+1,j}} = -i\sqrt{\frac{\eta_j}{\xi_{j+1,j}}} + i\sqrt{\frac{\eta}{\xi_{n,1}}}\frac{\partial \xi_{n,1}}{\partial \xi_{j+1,j}}, \qquad \text{for } 1 \le j \le n-1, \tag{D.3}$$

$$\frac{\partial p}{\partial t_{j+1,j}} = i\sqrt{\frac{\eta}{\xi_{n,1}}}\frac{\partial \xi_{n,1}}{\partial t_{j+1,j}}, \qquad \text{for } 2 \le j \le n-1. \tag{D.4}$$

The saddle-point is determined as follows. Equations (D.1), (D.2) imply

$$\sqrt{\frac{\xi_{n,1}^*}{\eta^*}} = (T_b)_{n,1} - i\left(L - \frac{2\pi\gamma}{s_*}\right), \qquad \sqrt{\frac{\xi_{j+1,j}^*}{\eta_j^*}} = (T_b)_{j+1,j}. \tag{D.5}$$

Meanwhile, (66) generalize to

$$\xi_{k+1,1} = \xi_{k+1,k} + \xi_{k,1} + 2\sqrt{\xi_{k+1,k}\xi_{k,1}}\cosh(t_{k+1,k} - t_{k,1}), \tag{D.6}$$

$$t_{k+1,1} = \frac{1}{2}\ln\left(\frac{\sqrt{\xi_{k+1,k}}e^{t_{k+1,k}} + \sqrt{\xi_{k,1}}e^{t_{k,1}}}{\sqrt{\xi_{k+1,k}}e^{-t_{k+1,k}} + \sqrt{\xi_{k,1}}e^{-t_{k,1}}}\right), \qquad \text{for } 2 \le k \le n-1. \tag{D.7}$$

We note

$$\frac{\partial \xi_{k+1,1}}{\partial t_{k+1,k}} = -\frac{\partial \xi_{k+1,1}}{\partial t_{k,1}} = 2\sqrt{\xi_{k+1,k}\xi_{k,1}}\sinh(t_{k+1,k} - t_{k,1}), \tag{D.8}$$

and for $1 \le j \le k-1$,

$$\boxed{\begin{aligned}\frac{\partial \xi_{k+1,1}}{\partial t_{j+1,j}} &= \frac{\partial \xi_{k+1,1}}{\partial \xi_{k,1}}\frac{\partial \xi_{k,1}}{\partial t_{j+1,j}} + \frac{\partial \xi_{k+1,1}}{\partial t_{k,1}}\frac{\partial t_{k,1}}{\partial t_{j+1,j}} \\ &= \sqrt{\frac{\xi_{k+1,1}}{\xi_{k,1}}}\cosh(t_{k+1,k} - t_{k,1})\frac{\partial \xi_{k,1}}{\partial t_{j+1,j}} - 2\sqrt{\xi_{k+1,k}\xi_{k,1}}\sinh(t_{k+1,k} - t_{k,1})\frac{\partial t_{k,1}}{\partial t_{j+1,j}}.\end{aligned}}$$

$$\tag{D.9}$$

In particular,

$$\frac{\partial \xi_{k+1,1}}{\partial t_{j+1,j}}\bigg|_{t_{k+1,k}=t_{k,1},\ldots,t_{j+2,j+1}=t_{j+1,1}} = \sqrt{\frac{\xi_{k+1,1}}{\xi_{j+1,1}}}\sqrt{\xi_{j+1,j}\xi_{j,1}}\,2\sinh\left(t_{j+1,j}-t_{j,1}\right), \quad \text{(D.10)}$$

for $1 \le j \le k$ by induction on $n = k-j \ge 0$. Then it follows from (D.4) and (D.10), and (D.6), (D.7) that

$$t^*_{j+1,j} = t^*_{j,1} = t_{21}, \quad \text{for } 2 \le j \le n-1, \qquad \sqrt{\xi^*_{n,1}} = \sum_{j=1}^{n-1}\sqrt{\xi^*_{j+1,j}}. \qquad \text{(D.11)}$$

These relations express the linearity of the motion of the particle in flat space, and can be used to easily evaluate derivatives at the saddle point at fixed $\{t_{j+1,j}\}$. For example,

$$\frac{\partial \xi_{n,1}}{\partial \xi_{j+1,j}}\bigg|_* = \sqrt{\frac{\xi^*_{n,1}}{\xi^*_{j+1,j}}}, \qquad \text{(D.12)}$$

so that (D.3) implies

$$\eta^* = \eta^*_1 = \cdots = \eta^*_{n-1}. \qquad \text{(D.13)}$$

Collecting (D.5), (D.11), and (D.13), the saddle-point is given by

$$s_* = s^*_1 = \cdots = s^*_{n-1} = \frac{2\pi}{\beta},$$

$$t^*_{j+1,j} = t_{21} \quad \text{for } 2 \le j \le n-1, \qquad \sqrt{\frac{\xi^*_{j+1,j}}{\eta^*}} = (T_b)_{j+1,j}, \quad \text{for } 1 \le j \le n-1. \qquad \text{(D.14)}$$

Let us now consider the expansion of the exponent to quadratic order about the saddle point. From (68) and (D.14), $p_* = -\frac{2\pi^2}{\beta}$. From (D.1)-(D.4) and (D.9), (D.11), we find the non-vanishing second derivatives are

$$a = \frac{\partial^2 p}{\partial s^2}\bigg|_* = \frac{2\pi}{s_*} - i\left(\frac{s_*}{\gamma}\right)^2\frac{1}{2\eta_*}\sqrt{\frac{\xi^*_{n,1}}{\eta^*}}, \qquad a_j = \frac{\partial^2 p}{\partial s_j^2}\bigg|_* = i\left(\frac{s_*}{\gamma}\right)^2\frac{1}{2\eta_*}\sqrt{\frac{\xi^*_{j+1,j}}{\eta^*}},$$

$$b_j = \frac{\partial^2 p}{\partial s\,\partial \xi_{j+1,j}}\bigg|_* = -\frac{\partial^2 p}{\partial s_j\,\partial \xi_{j+1,j}}\bigg|_* = -i\frac{s_*}{\gamma}\frac{1}{2\sqrt{\eta^*\xi^*_{j+1,j}}}, \qquad \text{for } 1 \le j \le n-1, \qquad \text{(D.15)}$$

and

$$c_{jk} = \frac{\partial^2 p}{\partial t_{j+1,j}\,\partial t_{k+1,k}}\bigg|_* = i\sqrt{\frac{\eta^*}{\xi^*_{n,1}}}\frac{\partial^2 \xi_{n,1}}{\partial t_{j+1,j}\,\partial t_{k+1,k}}\bigg|_*$$

$$= 2i\sqrt{\frac{\eta^*}{\xi^*_{n,1}}}\begin{cases}\sqrt{\xi^*_{j+1,j}}\left(\sqrt{\xi^*_{n,1}}-\sqrt{\xi^*_{j+1,j}}\right), & j = k, \\ -\sqrt{\xi^*_{j+1,j}\xi^*_{k+1,k}}, & j \ne k,\end{cases} \qquad \text{(D.16)}$$

for $2 \le j, k \le n-1$.

The calculation of (D.16) proceeds as follows. Let us show in sequence

$$\left.\frac{\partial t_{k+1,1}}{\partial t_{j+1,j}}\right|_* = \sqrt{\frac{\xi^*_{j+1,j}}{\xi^*_{k+1,1}}}, \qquad \text{for } j \leq k, \tag{D.17}$$

$$\left.\frac{\partial^2 \xi_{k+1,1}}{\partial t^2_{j+1,j}}\right|_* = 2\sqrt{\xi^*_{j+1,j}}\left(\sqrt{\xi^*_{k+1,1}} - \sqrt{\xi^*_{j+1,j}}\right), \qquad \text{for } j \leq k, \tag{D.18}$$

$$\left.\frac{\partial^2 \xi_{l+1,1}}{\partial t_{j+1,j} \partial t_{k+1,k}}\right|_* = -2\sqrt{\xi^*_{j+1,j}\xi^*_{k+1,k}}, \qquad \text{for } j,k \leq l, \; j \neq k, \tag{D.19}$$

by use of induction on $n = k - j$ or $n = l - k$, respectively.

- (D.17): The case $n = 0$ holds by taking the derivative of (D.7) with $k = 2$, with respect to $t_{32}$. Suppose (D.17) holds for some $n \geq 0$. Then for $j = k - n - 1$,

$$\left.\frac{\partial t_{k+1,1}}{\partial t_{j+1,j}}\right|_* = \left.\frac{\partial t_{k+1,1}}{\partial t_{k,1}}\frac{\partial t_{k,1}}{\partial t_{j+1,j}} + \frac{\partial t_{k+1,1}}{\partial \xi_{k,1}}\frac{\partial \xi_{k,1}}{\partial t_{j+1,j}}\right|_* = \sqrt{\frac{\xi^*_{k,1}}{\xi^*_{k+1,1}}}\sqrt{\frac{\xi^*_{j+1,j}}{\xi^*_{k,1}}} = \sqrt{\frac{\xi^*_{j+1,j}}{\xi^*_{k+1,1}}}. \tag{D.20}$$

- (D.18): The case $n = 0$ holds by (D.8). Suppose (D.18) holds for some $n \geq 0$. Then for $j = k - n - 1$, we have from (D.9), (D.10), and (D.11),

$$\begin{aligned}\left.\frac{\partial^2 \xi_{k+1}}{\partial t^2_{j+1,j}}\right|_* &= \left.\sqrt{\frac{\xi^*_{k+1,k}}{\xi^*_{k,1}}}\frac{\partial^2 \xi_{k,1}}{\partial t^2_{j+1,j}} + 2\sqrt{\xi_{k+1,k}\xi_{k,1}}\left(\frac{\partial t_{k,1}}{\partial t_{j+1,j}}\right)^2\right|_* \\ &= 2\sqrt{\xi^*_{j+1,j}}\left(\sqrt{\xi^*_{k+1,1}} - \sqrt{\xi^*_{j+1,j}}\right).\end{aligned} \tag{D.21}$$

- (D.19): Without loss of generality, let $j < k$. Consider $n = l - k = 0$. From (D.8),

$$\left.\frac{\partial^2 \xi_{l+1,1}}{\partial t_{k+1,k}\partial t_{j+1,j}}\right|_* = \left.-2\sqrt{\xi^*_{k+1,k}\xi^*_{k,1}}\frac{\partial t_{k,1}}{\partial t_{j+1,j}}\right|_* = -2\sqrt{\xi^*_{k+1,k}\xi^*_{j+1,j}}. \tag{D.22}$$

Now suppose (D.19) holds for some $n \geq 0$. Then for $k = l - n - 1$, $j < k$, we have from (D.9)

$$\begin{aligned}\left.\frac{\partial^2 \xi_{l+1,1}}{\partial t_{k+1,k}\partial t_{j+1,j}}\right|_* &= \left.\sqrt{\frac{\xi^*_{l+1,1}}{\xi^*_{l,1}}}\frac{\partial^2 \xi_{l,1}}{\partial t_{k+1,k}\partial t_{j+1,j}} + 2\sqrt{\xi^*_{l+1,l}\xi^*_{l,1}}\frac{\partial t_{l,1}}{\partial t_{k+1,k}}\frac{\partial t_{l,1}}{\partial t_{j+1,j}}\right|_* \\ &= -2\sqrt{\xi^*_{k+1,k}\xi^*_{j+1,j}}.\end{aligned} \tag{D.23}$$

From (D.18) and (D.19), we have

$$\left.\frac{\partial^2 \xi_{n,1}}{\partial t_{j+1,j}\partial t_{k+1,k}}\right|_* = \begin{cases} 2\sqrt{\xi^*_{j+1,j}}\left(\sqrt{\xi^*_{n,1}} - \sqrt{\xi^*_{j+1,j}}\right), & j = k, \\ -2\sqrt{\xi^*_{j+1,j}\xi^*_{k+1,k}}, & j \neq k, \end{cases} \tag{D.24}$$

and thus (D.16).

Denoting $x = s - s_*$, $x_j = s_j - s_j^*$, $y_j = \xi_{j+1,j} - \xi_{j+1,j}^*$, and $z_j = t_{j+1,j} - t_{j+1,j}^*$, we have found the exponent (68) has the expansion

$$
p - p_* \approx \frac{1}{2} a \left( x + \sum_{j=1}^{n-1} \frac{b_j y_j}{a} \right)^2 + \frac{1}{2} \sum_{j=1}^{n-1} a_j \left( x_j - \frac{b_j y_j}{a_j} \right)^2 + \frac{1}{2} \sum_{j=1}^{n-1} -b_j^2 \left( \frac{1}{a} + \frac{1}{a_j} \right) y_j^2
$$
$$
- \frac{1}{a} \sum_{1 \le j < k \le n-1} b_j b_k y_j y_k + \frac{1}{2} \sum_{j,k=2}^{n-1} z_j c_{jk} z_k \,. \tag{D.25}
$$

Taking the limit (64), in the leading saddle-point approximation we can neglect the cross terms in second derivatives of $y_j$ with coefficients $-\frac{b_j b_k}{a} \sim O\left( (\frac{s_*}{\gamma})^3 \right)$. We also note

$$
\det(c_{jk}) = \left( 2i \sqrt{\eta^*} \right)^{n-2} \sqrt{\frac{\prod_{j=1}^{n-1} \xi_{j+1,j}}{\xi_{n,1}}} \,. \tag{D.26}
$$

Then the leading saddle-point evaluation of (67) leads to the Markov property (71) with two-event probability distributions given by (65).

# E  Closed formulae for the saddle-point method

## E.1  Single-variable case

Campbell, Fröman, and Walles [26] gave a closed formula for the coefficents in the saddle-point expansion of an integral in a single variable. This is given as follows:

We consider an integral of the form

$$
I = \int_C dz \, f(z) e^{-p(z)} \,, \tag{E.1}
$$

where $p(z)$ is holomorphic and proportional to a large parameter, and there is a saddle point at $z = z_*$, $p'(z_*) = 0$. If the exponent function has first non-vanishing derivative

$$
p(z) = p(z_*) + \frac{1}{\mu!} p^{(\mu)}(z_*)(z - z_*)^\mu + \cdots \,, \tag{E.2}
$$

and the integration contour $C$ approaches and leaves the saddle-point at steepest-descent angles

$$
\theta_l = \frac{-\arg\left( p_*^{(\mu)} \right) + 2\pi l}{\mu} \qquad (l \in \mathbb{Z}) \,, \tag{E.3}
$$

with $l = k_1$ and $l = k_2$, respectively,

$$
\boxed{
\begin{aligned}
I &= e^{-p_*} \left\{ \sum_{s=0}^{S-1} \Gamma\left( \frac{s+1}{\mu} \right) \alpha_s \left( e^{2\pi i k_2 (s+1)/\mu} - e^{2\pi i k_1 (s+1)/\mu} \right) + O\left( p^{-(s+1)/\mu} \right) \right\} \,, \\
\alpha_s &= \frac{1}{\mu s!} \frac{d^s}{dz^s} \left\{ f(z) \left( \frac{p(z) - p(z_*)}{(z - z_*)^\mu} \right)^{-(s+1)/\mu} \right\} \Bigg|_{z=z_*} \,.
\end{aligned}
} \tag{E.4}
$$

Using the partial ordinary Bell polynomials

$$
\left( p_1 x + p_2 x^2 + p_3 x^3 + \cdots \right)^j = \sum_{i=j}^{\infty} \hat{B}_{i,j}(p_1, p_2, p_3, \cdots) x^i \,, \tag{E.5}
$$

and expansion of functions

$$\frac{p(z) - p(z_*)}{(z - z_*)^\mu} = \sum_{k=0}^\infty p_k (z - z_*)^k, \qquad f(z) = \sum_{k=0}^\infty f_k (z - z_*)^k, \tag{E.6}$$

one has

$$\alpha_S = \frac{1}{\mu} p_0^{-(s+1)/\mu} \sum_{i=0}^s f_{s-i} \sum_{j=0}^i p_0^{-j} \binom{-(s+1)/\mu}{j} \hat{B}_{i,j}(p_1, \dots, p_{i-j+1}). \tag{E.7}$$

Specializing to the case of $\mu = 2$ ($k_1 = -1, k_2 = 0$),

$$I = e^{-p(z_*)} \sum_{m=0}^\infty \Gamma\left(m + \frac{1}{2}\right) p_0^{-(m+\frac{1}{2})} \sum_{i=0}^{2m} f_{2m-i} \sum_{j=0}^i p_0^{-j} \binom{-(m+\frac{1}{2})}{j} \hat{B}_{i,j}(p_1, \dots, p_{i-j+1}). \tag{E.8}$$

## E.2 Multi-variable case

Here we derive a closed formula for the coefficients of the saddle-point expansion of a multi-variable integral. To our knowledge, this result has not previously appeared in the literature.

Let $\mathcal{N}$ be the number of variables. We use the notation that for $\boldsymbol{i} \in \mathbb{N}^{\mathcal{N}}, \boldsymbol{z} \in \mathbb{C}^{\mathcal{N}}$,

$$\boldsymbol{i}! = \prod_{\mathcal{M}=1}^{\mathcal{N}} i_\mathcal{M}!, \quad |\boldsymbol{i}| = \sum_{\mathcal{M}=1}^{\mathcal{N}} i_\mathcal{M}, \quad \boldsymbol{z}^{\boldsymbol{i}} = \prod_{\mathcal{M}=1}^{\mathcal{N}} z_\mathcal{M}^{i_\mathcal{M}}, \tag{E.9}$$

and $\boldsymbol{0}$, $\boldsymbol{1}$ are the vectors of 0's and 1's, respectively, in $\mathbb{N}^{\mathcal{N}}$. We will also use multivariate Bell polynomials $\hat{B}_{ij}(x)$, $x : \mathbb{N}^{\mathcal{N}} \to \mathbb{C}$ [22] whose definition we gave above (76).

We seek to evaluate

$$I = \int d\boldsymbol{z}\, f(\boldsymbol{z}) e^{-p(\boldsymbol{z})}, \tag{E.10}$$

the multivariate generalization of (E.1). Let as assume that the matrix of second derivatives $\frac{\partial^2 p}{\partial z_\mathcal{L} \partial z_\mathcal{M}}$ at the saddle point is non-degenerate and diagonalizable. (It is clear our formula below can be generalized to analogous cases where the first non-vanishing derivatives appear at order $\mu \geq 3$.) Then let us expand

$$p(\boldsymbol{z}) - p(\boldsymbol{z}_*) = \sum_{\mathcal{M}=1}^{\mathcal{N}} p_\mathcal{M} (z_\mathcal{M} - z_\mathcal{M}^*)^2 + \sum_{|\boldsymbol{k}|=3}^\infty p_{\boldsymbol{k}} (\boldsymbol{z} - \boldsymbol{z}_*)^{\boldsymbol{k}}. \tag{E.11}$$

The main idea of our derivation is to expand the exponential

$$\exp\left(-\sum_{|\boldsymbol{k}|=3}^\infty p_{\boldsymbol{k}} (\boldsymbol{z} - \boldsymbol{z}_*)^{\boldsymbol{k}}\right) = \sum_{j=0}^\infty \frac{(-1)^j}{j!} \left(\sum_{|\boldsymbol{k}|=3}^\infty p_{\boldsymbol{k}} (\boldsymbol{z} - \boldsymbol{z}_*)^{\boldsymbol{k}}\right)^j = \sum_{j=0}^\infty \frac{(-1)^j}{j!} \sum_{|\boldsymbol{i}|=3j}^\infty \hat{B}_{\boldsymbol{i}j}(p) (\boldsymbol{z} - \boldsymbol{z}_*)^{\boldsymbol{i}}. \tag{E.12}$$

When the integral (E.10) is performed, higher-order terms in the series are further suppressed in large $p$ because they carry sufficiently more powers of $z_\mathcal{M} - z_\mathcal{M}^*$.

Also expanding

$$f(\boldsymbol{z}) = \sum_{|\boldsymbol{k}|=0}^\infty f_{\boldsymbol{k}} (\boldsymbol{z} - \boldsymbol{z}_*)^{\boldsymbol{k}}, \tag{E.13}$$

and using

$$\int_{-\infty}^{\infty} dz\, e^{-p(z-z_*)^2}(z-z_*)^n = \frac{1}{2}(1+(-1)^n)\, p^{-(n+1)/2}\Gamma\left(\frac{n+1}{2}\right), \qquad \text{for } \text{Re}\, p > 0,\ n \in \mathbb{Z}, \tag{E.14}$$

then collecting terms with fixed powers of $p$ in (E.10),

$$p^{-\left(m+\frac{\mathcal{N}}{2}\right)}, \quad m \in \mathbb{N}, \qquad \text{with} \qquad |\boldsymbol{k}| + |\boldsymbol{i}| = 2(j+m), \tag{E.15}$$

we arrive at the final formula

$$\boxed{\begin{aligned}
I &= e^{-p_*} \sum_{m=0}^{\infty} \sum_{j=0}^{2m} \sum_{|\boldsymbol{i}|=3j}^{2(m+j)} \frac{(-1)^j}{j!} \hat{B}_{ij}(p) \\
&\quad \times \sum_{|\boldsymbol{k}|=2(m+j)-|\boldsymbol{i}|} f_{\boldsymbol{k}} \prod_{\mathcal{M}=1}^{\mathcal{N}} \frac{1}{2}\left(1+(-1)^{(k+i)_{\mathcal{M}}}\right) p_{\mathcal{M}}^{-((k+i)_{\mathcal{M}}+1)/2} \Gamma\left(\frac{(k+i)_{\mathcal{M}}+1}{2}\right).
\end{aligned}} \tag{E.16}$$

In the case $\mathcal{N} = 1$, we can verify our formula reproduces that of Campbell, Fröman, and Walles, (E.8). Writing $i = i' + 2j$ and noting $\hat{B}_{i'+2j,j}(0,0,p_1,p_2,\dots) = \hat{B}_{i',j}(p_1,p_2,\dots)$,

$$\begin{aligned}
I &= e^{-p_*} \sum_{m=0}^{\infty} \sum_{i'=0}^{2m} \sum_{j=0}^{i'} f_{2m-i'} \frac{(-1)^j}{j!} \hat{B}_{i',j}(p_1,p_2,\dots)\, p_0^{-\left(m+\frac{1}{2}\right)-j} \Gamma\left(m+\frac{1}{2}+j\right) \\
&= e^{-p_*} \sum_{m=0}^{\infty} p_0^{-\left(m+\frac{1}{2}\right)} \Gamma\left(m+\frac{1}{2}\right) \sum_{i'=0}^{2m} f_{2m-i'} \sum_{j=0}^{i'} p_0^{-j} \binom{-\left(m+\frac{1}{2}\right)}{j} \hat{B}_{i',j}(p_1,\dots,p_{i'-j+1}). \tag{E.17}
\end{aligned}$$

## F  Evaluation of generator equation

Here we give details of the evaluation of the generator equation (5) for the asymptotic quantum stochastic process corresponding to flat JT gravity, extracted from the local limit of the quantum theory of the boundary of AdS JT gravity.

### F.1  Two-event integrals

Let us consider the two-event integral appearing on the right-hand-side of the equation,

$$\int \mathcal{D}x_1 \frac{q_{T_2,T_1}(x_3,x_1)}{q_{T_1}(x_1)} \Phi(x_1) = \int \mathcal{D}x_1 \frac{\langle x_3|e^{-iHT_{21}}\rho|x_1\rangle \langle x_1|e^{iHT_{21}}|x_3\rangle}{\langle x_1|\rho|x_1\rangle} \Phi(x_1). \tag{F.1}$$

We use the inverse of the factor $\langle x_1|\rho|x_1\rangle = 2\text{vol}(\text{H}(x_1))$ to fix the time-like coordinate of the point at time $T_2$ relative to that at $T_1$, $\text{vol}(\text{H}(x_1))^{-1} = \delta(t_{31} - t)$. See discussion near (33). Using the local propagator (57) in the two-point functions (22), (27) and the semi-classical approximations (59), we have that in the holographic and semi-classical limit the integral localizes to regions $(x_3; x_1) \in$ region 3,4 and becomes

$$\begin{aligned}
&\frac{1}{2} \int_{3,4} \frac{d\xi}{\gamma} \int ds\, s\, \frac{e^{-(\beta+iT)s^2/2+2\pi s}}{Z(2\pi)^2} \frac{\sqrt{\pi}}{(\eta\xi)^{1/4}} \left\{ e^{-\frac{1}{4}\pi i - 2i\sqrt{\eta\xi}} \left(1 + \frac{i}{16}\frac{1}{\sqrt{\eta\xi}}\right) + \text{c.c.} \right\} \\
&\quad \times \int ds'\, s'\, \frac{e^{iTs'^2/2}}{(2\pi)^2} \frac{\sqrt{\pi}}{(\eta'\xi)^{1/4}} \left\{ e^{-\frac{1}{4}\pi i - 2i\sqrt{\eta'\xi}} \left(1 + \frac{i}{16}\frac{1}{\sqrt{\eta'\xi}}\right) + \text{c.c.} \right\} \Phi(\xi,t), \tag{F.2}
\end{aligned}$$

where we have used the shorthand $T_{21} = T$, $\xi_{13} = \xi_{31} = \xi$.

To apply the formula for saddle-point expansion (75), we consider the expansion of the exponent about its saddle-point (63). To make the exponent analytic, we change variables as

$$l = \xi^{1/2}, \tag{F.3}$$

after which its second-order expansion takes the form

$$p - p_* \approx \frac{1}{2} a \left( x + \frac{b}{a} y \right)^2 + \frac{1}{2} a' \left( x' - \frac{b}{a'} y \right)^2 + \frac{1}{2} \left( -b^2 \left( \frac{1}{a} + \frac{1}{a'} \right) \right) y^2, \tag{F.4}$$

where

$$a = \left. \frac{\partial^2 p}{\partial s^2} \right|_*, \; a' = \left. \frac{\partial^2 p}{\partial s'^2} \right|_*, \; b = \left. \frac{\partial^2 p}{\partial s \partial l} \right|_* = - \left. \frac{\partial^2 p}{\partial s' \partial l} \right|_*, \qquad x = s - s_*, \; x' = x' - x'_*, \; y = l - l_*. \tag{F.5}$$

To make the matrix of second derivatives diagonal, we should change variables in the integral as

$$s = w - \frac{b}{a} l, \quad s' = w' + \frac{b}{a'} l, \qquad l. \tag{F.6}$$

Then the result of applying (75) is given by

$$\int \mathcal{D} x_1 \frac{q_{T_2, T_1}(x_3, x_1)}{q_{T_1}(x_1)} \Phi(x_1) = \sum_{(x_3; x_1) \in \text{regions } 3,4} \frac{1}{2} \left( 1 + \left( \frac{i}{4\sqrt{\eta}} - \frac{(T_b)_{21}}{8\beta \eta^{3/2}} \left( 1 + \frac{4\eta}{\gamma} \right) \right) \partial_{l_{31}} \right. \\ \left. + \left( \frac{i}{4} + \frac{s^2}{8\beta \eta \gamma^2} (T_b)_{21} \right) (T_b)_{21} \partial^2_{l_{31}} + O\left( \gamma^{-2} \right) \right) \Phi(l_{31}, t) \Big|_*. \tag{F.7}$$

Similarly, we can evaluate the two-event integral on the LHS as

$$\lim_{T_{21} \to 0^+} \int \mathcal{D} x_1 \frac{\partial_{T_2} q_{T_2, T_1}(x_3, x_1)}{q_{T_1}(x_1)} \Phi(x_1)$$
$$= \sum_{(x_3; x_1) \in \text{regions } 3,4} \lim_{x_1 \to x_3} \frac{\gamma}{2} \left( \sqrt{\eta} \partial_{l_{31}} - \frac{1}{8\beta \eta^{3/2}} \left( 1 + \frac{4\eta}{\gamma} \right) \partial_{l_{31}} + \frac{i}{2} \partial^2_{l_{31}} + O\left( \gamma^{-2} \right) \right) \Phi(l_{31}, t) \Big|_*. \tag{F.8}$$

Note we only need to compute to $O(\gamma^{-1})$ to match the computation of integrals to $O(\gamma^{-2})$ on right-hand-side, as the latter are divided by $T_{32} = (T_b)_{32}/\gamma \sim O(\gamma^{-1})$. The action of $\partial_{T_2}$ brings down a factor of $E' - E = (s'^2 - s^2)/2$ in (F.2), so the integral is enhanced by a factor of $\gamma^2$ in semi-classical counting. Thus we expand leading terms in the amplitude in (F.2) to order $m = 2$ in the formula (75), and subleading terms, to order $m = 1$—sub-subleading terms do not contribute as the $m = 0$ evaluation right at the saddle-point is identically zero due to $s_* = s'_*$.

## F.2 Three-event integrals

Here we consider the three-event integral

$$\int \mathcal{D} x_1 \mathcal{D} x_2 \frac{\langle x_3 | e^{-iH T_{31}} \rho | x_1 \rangle \langle x_1 | e^{iH T_{21}} | x_2 \rangle \langle x_2 | e^{iH T_{32}} | x_3 \rangle}{\langle x_1 | \rho | x_1 \rangle} \sum_{|k|=0}^{\infty} \frac{\Phi^{(k)}(x_{13} = x_{12})}{k!} (x_2 - x_3)^k. \tag{F.9}$$

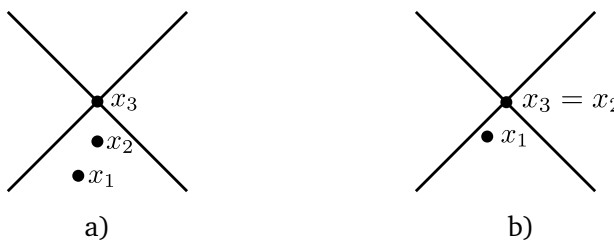

Figure 11: In the Taylor expansion of $\Phi(x_1)$, we expand the configuration a) about b).

Note in order to be consistent with our evaluation of two-event integrals above, we should factor out $\langle x_1|\rho|x_1\rangle = 2\text{vol}(\mathrm{H}(x_1))$ as $\text{vol}(\mathrm{H}(x_1))^{-1} = \delta(t_{21} - t)$.

In expanding $\Phi(x_1)$, we have chosen to specify the position of $\Phi$ relative to the fixed point $x_3$, and expand $x_{13}$ about $x_{12}$; see Figure 11. In evaluating the integral, we perform this expansion in relative coordinates as follows. In the semi-classically relevant regions $(x_3; x_2), (x_2; x_1) \in$ region 3, $(x_3; x_2), (x_2; x_1) \in$ region 4, we have the composition formulas (66). Then using absolute coordinates $(u, v)$ related to relative coordinates as (58), we can solve for displacements $\Delta x_1 = x_1 - \bar{x}_1$ such that $x_{3\bar{1}} = x_{12}$:

$$\Delta u_1 = \pm\left(l_{12}e^{-t_{12}} - l_{31}e^{-t_{31}}\right), \qquad \Delta v_1 = \pm\left(l_{12}e^{t_{12}} - l_{31}e^{t_{31}}\right), \tag{F.10}$$

with upper (lower) signs for the case $(x_3; x_2), (x_2; x_1) \in$ region 3 (region 4). With $x_3$ being fixed, we can convert this displacement to $\Delta x_{31} = \left.\dfrac{\partial x_{31}}{\partial x_1^\mu}\right|_{x_1=\bar{x}_1}\Delta x_1^\mu + \dfrac{1}{2!}\left.\dfrac{\partial^2 x_{31}}{\partial x_1^\mu \partial x_1^\nu}\right|_{x_1=\bar{x}_1}\Delta x_1^\mu \Delta x_1^\nu + O\left(\Delta x_1^3\right)$, and find

$$\Delta l_{13} = l_{13}\cosh(t_{13} - t_{12}) - l_{12} - \frac{1}{2}\frac{l_{13}^2}{l_{12}}\sinh^2(t_{13} - t_{12}) + O\left(\Delta x_1^3\right),$$

$$\Delta t_{13} = \frac{l_{13}}{l_{12}}\sinh(t_{13} - t_{12}) - \frac{l_{13}}{l_{12}^2}(l_{13}\cosh(t_{13} - t_{12}) - l_{12})\sinh(t_{13} - t_{12}) + O\left(\Delta x_1^3\right), \tag{F.11}$$

which we can substitute in the Taylor expansion

$$\Phi(l_{31}, u_{31}) = \left(\Phi + \Delta l_{31}\Phi^{(1,0)} + \Delta t_{31}\Phi^{(0,1)} + \frac{1}{2}\Delta l_{31}^2\Phi^{(2,0)}\right.$$
$$\left. + \Delta l_{31}\Delta t_{31}\Phi^{(1,1)} + \frac{1}{2}\Delta l_{31}^2\Phi^{(2,0)} + \cdots\right)(l_{21}, t_{21}). \tag{F.12}$$

Let us now go through the saddle-point expansion of (F.9). Proceeding via similar steps as in (F.2), the integral evaluates in the semi-classical limit to

$$\frac{e^{\frac{1}{4}\pi i}}{32\pi^{9/2}Z\gamma^2}\int_{\substack{(x_1;x_2),(x_2;x_3)\in\text{region 3}\\ \in\text{region 4}}} dl_{12}dt_{12}\int dl_{23}dt_{23}\,\delta(t_{12} - t)\int ds\,ds_1\,ds_2\frac{ss_1s_2}{(\eta\eta_1\eta_2)^{1/4}}\frac{l_{12}^{1/2}l_{23}^{1/2}}{l_{13}^{1/2}}$$

$$\times\left(1 + \frac{i}{16}\left(\frac{1}{\sqrt{\eta}l_{13}} - \frac{1}{\sqrt{\eta_1}l_{12}} - \frac{1}{\sqrt{\eta_2}l_{23}}\right) + O\left(\gamma^{-2}\right)\right)F(l_{12}, t_{12}, l_{23}, t_{23})e^{-p(s,s_1,s_2,l_{12},t_{12},l_{23},t_{23})}, \tag{F.13}$$

where the exponent $p$ is given by (68) with $n = 3$ and the function $F$ stands in for the expansion given by (F.12), (F.11). Note we use composition formulas for $l_{31}, t_{31}$ given by (66). The

second-order expansion of the exponent about its linear saddle (70) takes the form

$$p - p_* \approx \frac{1}{2} a \left( x + \frac{b}{a} \sum_{j=1}^{2} y_j \right)^2 + \frac{1}{2} \sum_{j=1}^{2} a_j \left( x_j - \frac{b}{a_j} y_j \right)^2 + \frac{1}{2} \sum_{j=1}^{2} \left( -b^2 \left( \frac{1}{a} + \frac{1}{a_j} \right) \right) y_j^2$$
$$- \frac{1}{a} b^2 y_1 y_2 + \frac{1}{2} c z^2 , \tag{F.14}$$

where

$$a = \left. \frac{\partial^2 p}{\partial s^2} \right|_* , \quad a_j = \left. \frac{\partial^2 p}{\partial s_j^2} \right|_* , \quad b = \left. \frac{\partial^2 p}{\partial s \partial l_{j,j+1}} \right|_* = - \left. \frac{-\partial^2 p}{\partial s_j \partial l_{j,j+1}} \right|_* , \quad c = \left. \frac{\partial^2 p}{\partial t_{23}^2} \right|_* , \tag{F.15}$$
$$x = s - s_* , \quad x_j = s_j - s_j^* , \quad y_j = l_{j,j+1} - l_{j,j+1}^* , \quad z = t_{23} - t_{23}^* , \quad j, k = 1, 2 .$$

To diagonalize the expansion, we change variables to

$$w = s + \frac{b}{a} \sum_{j=1}^{2} l_{j,j+1} , \qquad w_j = s_j - \frac{b}{a_j} l_{j,j+1} . \tag{F.16}$$

Further, we diagonalize the matrix of derivatives w.r.t. $l_{j,j+1}$ by working with rotated variables

$$v_+ = l_{12} - X l_{23} , \qquad v_- = X l_{12} + l_{23} , \tag{F.17}$$

$$X = -\frac{a(a_1 - a_2)}{2 a_1 a_2} \left( 1 - \sqrt{1 + \frac{4 a_1^2 a_2^2}{a^2 (a_1 - a_2)^2}} \right) \approx \frac{a_1 a_2}{a(a_1 - a_2)} \sim O\left( \left( \frac{s}{\gamma} \right)^3 \right) . \tag{F.18}$$

The eigenvalues of the matrix are given by

$$\alpha_\pm = \frac{2 a_1 a_2 + a(a_1 + a_2) \mp a(a_1 - a_2) \sqrt{1 + \frac{4 a_1^2 a_2^2}{a^2 (a_1 - a_2)^2}}}{2 a a_1 a_2} \approx \frac{1}{a} + \frac{1}{a_{1,2}} \mp \frac{a_1 a_2}{a^2 (a_1 - a_2)} , \tag{F.19}$$

and the exponent expands quadratically with respect to new variables (F.17) as

$$p - p_* \sim -\frac{b^2}{2} \left( \frac{1}{1 + X^2} \right) \left( \alpha_+ \left( v_+ - v_+^* \right)^2 + \alpha_- \left( v_- - v_-^* \right)^2 \right) . \tag{F.20}$$

Then the results of using (75) to evaluate terms of order zero, one, and two derivatives of the Taylor expansion (F.12) inside (F.9) are as follows:

$$\int \mathcal{D} x_1 \mathcal{D} x_2 \frac{q_{T_3, T_2, T_1}(x_3, x_2, x_1)}{q_{T_1}(x_1)} \Phi(x_{31} = x_{21})$$
$$= \sum_{(x_2; x_1) \in \text{regions } 3,4} \frac{1}{2} \left( 1 + \left( \frac{i}{4 \sqrt{\eta}} - \frac{(T_b)_{21}}{8 \beta \eta^{3/2}} \left( 1 + \frac{4 \eta}{\gamma} \right) \right) \partial_{l_{21}} \tag{F.21}$$
$$+ \left( \frac{i}{4} + \frac{s^2}{8 \beta \eta \gamma^2} (T_b)_{21} \right) (T_b)_{21} \partial_{l_{21}}^2 + O\left( (T_b)_{32}^2 \right) + O\left( \gamma^{-2} \right) \right) \Phi(l_{21}, t) \Big|_* ,$$

$$\int \mathcal{D}x_1 \mathcal{D}x_2 \frac{q_{T_3,T_2,T_1}(x_3,x_2,x_1)}{q_{T_1}(x_1)} \sum_{|\boldsymbol{k}|=1} \frac{\Phi^{(k)}(x_{31}=x_{21})}{\boldsymbol{k}!} \Delta x_{31}^{\boldsymbol{k}}$$
$$= \sum_{(x_2;x_1)\in \text{ regions } 3,4} \frac{1}{2}\Bigg( (T_b)_{32} \Bigg( \sqrt{\eta} \partial_{l_{21}} + \frac{i}{4} \partial_{l_{21}}^2 - \frac{(1+4\eta/\gamma)}{8\beta\eta^{3/2}} \partial_{l_{21}} \Bigg)$$
$$+ O\Big( (T_b)_{32}^2, (T_b)_{32}(T_b)_{21} \Big) + O\big(\gamma^{-2}\big) \Bigg) \Phi(l_{21},t) \Bigg|_* , \tag{F.22}$$

and

$$\int \mathcal{D}x_1 \mathcal{D}x_2 \frac{q_{T_3,T_2,T_1}(x_3,x_2,x_1)}{q_{T_1}(x_1)} \sum_{|\boldsymbol{k}|=2} \frac{\Phi^{(k)}(x_{31}=x_{21})}{\boldsymbol{k}!} \Delta x_{31}^{\boldsymbol{k}}$$
$$= \sum_{(x_2;x_1)\in \text{ regions } 3,4} \frac{1}{2}\Bigg( (T_b)_{32} \Bigg( \frac{-i}{4l_{21}^2} \partial_{t_{21}}^2 + \frac{i}{4} \partial_{l_{21}}^2 \Bigg) + + O\Big( (T_b)_{32}^2 \Big) + O\big(\gamma^{-2}\big) \Bigg) \Phi(l_{21},t) \Bigg|_* , \tag{F.23}$$

To $O\big(\gamma^{-1}\big)$, terms of order three or higher in the Taylor expansion of $\Phi(x_1)$ only give terms of $O\big(T_{32}^2\big)$, so do not contribute to the generator equation. Collecting the integrals (F.7), (F.8), (F.21), (F.22), (F.23), we note (F.7) and (F.21) cancel each other exactly, and that at first non-vanishing order in large $\gamma$, the generator equation reduces to (77).

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
