# Peer review of "Probabilistic deconstruction of a theory of gravity, Part I: flat space"

_SciPost Physics, doi:SciPost Phys. 15, 174 (2023)_

## Round 2 · Referee Report · Anonymous (Referee 1) · 2023-6-27

Strengths

1. Novel idea for deriving Einstein equations of motion in a controlled set-up
2. Tries to bridge notions in classical probability theory and holography

Weaknesses

1. Unclear how the techniques can generalize beyond 1+1 dimensions.
2. Various arguments of the paper were very difficult to follow.

Report

This paper tries to give a novel perspective on how bulk Einstein equations emerges via some quantum equation for the generator for a Markov process, in the context of the SYK-JT gravity correspondence.
While the ideas are novel, I found the discussion quite hard to follow at various places and the logic unclear. After the issues mentioned below are addressed, the paper could meet the criteria for SciPost Phys.

Requested changes

It would be helpful if the author expanded on the following questions, where possible:

1. The main player is the boundary action or the generalized Gibbons-Hawking term in equation 1. This is the same irrespective of a cosmological constant and arises on imposing Dirichlet boundary conditions on the metric variation. What is the role of the Dirichlet boundary condition from the microscopic perspective? How will things change if one considers Neumann boundary conditions where the boundary term is no longer the Gibbons-Hawking term? Will the arguments in this paper generalize in that case?

2. The worldline action of a point particle moving on the boundary of AdS2 spacetime is given by equation 15. The microscopic theory which involves the Majorana fermions in the SYK description has been replaced by a low energy effective quantum mechanics of a single point particle. It would appear that the information about the AdS background is already put in equation 15. If this is the case, then what does it mean for the bulk to be emerging? Is the answer to the following question known: Given Gibbons-Hawking as the boundary term in d-dimensions, can we identify the correct diffeomorphism invariant bulk term for which the Gibbons-Hawking term is the boundary term? The reason this question is of some relevance is that it is not clear from the discussion in the paper if AdS2 bulk emerges or has been put in by hand. Since some of the discussions later on considers a particular limit where only short time dynamics and hence flat space is of relevance, what role does the AdS/CFT correspondence play in this discussion apart from being a by-stander?

3. A technical question: Below equation 31, the author mentions that starting with eq 30, it is not possible to find a double expansion of a certain kind. Why is this not possible?

4. The large N (large gamma) limit appears to be crucial in derivation of the Markovian property. Is this related to large N factorisation of correlation functions in the AdS/CFT correspondence?

5. Forgetting about the underlying microscopic theory for a while [which appears to play very little role in the discussion in this paper], one could consider other solutions to Einstein equations (with matter) where instead of AdS2 one gets Lifshitz backgrounds. The derivation of equation 80 in the short time limit seems strange in this sense since even for such backgrounds, one would presumably get the same equation. Is this expected? Or would the stochastic averaging always give maximally symmetric spacetime?

6. Previously in the literature, people have talked about deriving Einstein equations from entanglement entropy. See eg. 1312.7856 Can the author comment on any potential connection with this line of research?

  • validity: -
  • significance: -
  • originality: -
  • clarity: -
  • formatting: -
  • grammar: -

Author:  Josephine Suh  on 2023-07-29  [id 3851]

(in reply to Report 1 on 2023-06-27)
Category:
remark
answer to question

We thank the referee for helpful questions, and hope our answers below can be satisfactory. We plan to include more quantitative expositions as appropriate in the revision of our manuscript.

First, some background regarding the relationship between microscopic SYK and the JT action:

The bulk $AdS_2$ spacetime emerges from the low-energy dynamics of the microscopic SYK model, with the coordinates of the bulk spacetime being directly related to form of the fermionic Green's function in SYK. In short, the form of the Green's function at low-energies ("soft mode") and the effective action governing the emergent mode ("Schwarzian") implies a description of the system in terms of a free matter field in $AdS_2$ interacting with a specific mode propagating between bulk points.

(The new point of view we are contributing in this paper with regards to this "emergent" spacetime is that we can understand it operationally and quantum-mechanically as the target space of a quantum observable, $i.e.$ the one resulting from quantizing the soft mode ("boundary" d.o.f.) . We then show the utility of this viewpoint by deriving the JT equivalent of Einstein's equations ("bulk" equations) using only the quantum mechanics resulting from quantizing the boundary mode.)

Now, the emergent Schwarzian action can be derived from the SYK model when one works in the $N \gg \beta J \gg 1$ limit ("holographic limit") and importantly, at long time scales $|\tau_1-\tau_2| J \gg 1$. The JT action in eq. 1 can be viewed as a completion to short time-scales of the Schwarzian action. The JT action, with Dirichlet boundary conditions and negative cosmological constant, reduces to the Schwarzian action at long time-scales. (Intuitively, the boundary particle resulting from quantizing the JT action in eq. 1 lives in exact $AdS_2$, whereas in the Schwarzian limit it lives in an asymptotic geometry near the boundary of $AdS_2$.)

Now, to address the referee's individual questions:

  1. The main player is the boundary action or the generalized Gibbons-Hawking term in equation 1. This is the same irrespective of a cosmological constant and arises on imposing Dirichlet boundary conditions on the metric variation. What is the role of the Dirichlet boundary condition from the microscopic perspective?

For the JT action to reduce to the Schwarzian action which describes an emergent mode in the SYK model, it is both true that i) the Dirichlet boundary condition should be imposed, and ii) that the cosmological constant should be set to be negative. (The negative cosmological constant enters because an explicit expression for extrinsic curvature $K$ depends on the curvature of the spacetime.)

How will things change if one considers Neumann boundary conditions where the boundary term is no longer the Gibbons-Hawking term? Will the arguments in this paper generalize in that case?

Our arguments will not generalize. Without Dirichlet boundary conditions, we cannot view the boundary action of JT as simply describing a particle propagating in background spacetime, and proceed with quantization, etc.

  1. The worldline action of a point particle moving on the boundary of AdS2 spacetime is given by equation 15. The microscopic theory which involves the Majorana fermions in the SYK description has been replaced by a low energy effective quantum mechanics of a single point particle. It would appear that the information about the AdS background is already put in equation 15.

See preliminary exposition. For the action in eq. 15 to describe the short-time completion of the emergent dynamics of SYK, it must be put on an AdS background.

If this is the case, then what does it mean for the bulk to be emerging?

What we are attempting to do is identify the direct operational meaning of the emergent spacetime and its volume measure relative to the quantum mechanics describing the boundary d.o.f. We argue that one should view "emergent spacetime" as the target space of a quantum observable, and the "volume measure" at a point of the emergent spacetime, as a probability measure constrained by the dynamics of the said observable.

Perhaps another way to put it is this: the bulk doesn't mysteriously "emerge", it exists as the target space of an (emergent) observable of a quantum mechanical system.

Is the answer to the following question known: Given Gibbons-Hawking as the boundary term in d-dimensions, can we identify the correct diffeomorphism invariant bulk term for which the Gibbons-Hawking term is the boundary term? The reason this question is of some relevance is that it is not clear from the discussion in the paper if AdS2 bulk emerges or has been put in by hand.

The $AdS_2$ geometry is a consequence of the low-energy dynamics of the SYK model. What we do in this paper is identify the direct operational meaning of this geometry relative to the quantum mechanics resulting from quantizing said dynamics.

Since some of the discussions later on considers a particular limit where only short time dynamics and hence flat space is of relevance, what role does the AdS/CFT correspondence play in this discussion apart from being a by-stander?

As we take the short-time limit in joint quantum distributions in eq. (4) to go to flat-space, the quantities retain knowledge of the density matrix that we started out with, i.e. the spectral density of the JT gravity in AdS. In short, AdS/CFT correspondence gave us a quantum system producing certain expectation values which one could one then deform to reach the flat-space limit.

Also, it is only in the limit eq. (16) or equivalently, eq. (29), the analogue of large $N$, large $\lambda$ in higher-dimensional AdS/CFT , that we can renormalize the action (15). Furthermore, all of our calculations of the quantum theory are done in the same limit, and our derivation of the bulk Einstein's equations from boundary quantum theory would not occur outside the limit.

  1. A technical question: Below equation 31, the author mentions that starting with eq 30, it is not possible to find a double expansion of a certain kind. Why is this not possible?

We found that such a double expansion does not exist in closed form, $i.e.$ the coefficients do not come in closed-form functions.

  1. The large N (large gamma) limit appears to be crucial in derivation of the Markovian property. Is this related to large N factorisation of correlation functions in the AdS/CFT correspondence?

The large $N$, $E \ll N$ limit in eq. (29) (analogue of large $N$, large $\lambda$ in higher-dimensional AdS/CFT) as well as the semi-classical limit $E \gg 1$ in eq. (31) are all crucial to the derivation of the Markovian property. A final ingredient is the short-distance or flat-space limit. Without the latter, we suspect that the Markovian property only holds in a "local" sense.

  1. Forgetting about the underlying microscopic theory for a while [which appears to play very little role in the discussion in this paper], one could consider other solutions to Einstein equations (with matter) where instead of AdS2 one gets Lifshitz backgrounds. The derivation of equation 80 in the short time limit seems strange in this sense since even for such backgrounds, one would presumably get the same equation. Is this expected? Or would the stochastic averaging always give maximally symmetric spacetime?

I believe the Lifshitz asymmetry will show up when going from the LHS to the RHS. The coordinates l, t on the LHS are relative coordinates between two point measuring geodesic distance and direction.

  1. Previously in the literature, people have talked about deriving Einstein equations from entanglement entropy. See eg. 1312.7856 Can the author comment on any potential connection with this line of research?

In technical terms, the work in eg. 1312.7856 crucially relied on an expansion around the CFT vacuum and also had the limitation of obtaining only the linearized Einstein's equations. Conceptually, entanglement entropy only characterizes the quantum state, and it is natural to expect that the dynamics of the boundary quantum system should be accounted for in order to fully reproduce Einstein's equations. In our work we are proposing to quantify the necessary quantum dynamics using joint quantum distributions in eq.(4).

---

## Round 2 · Referee Report · Anonymous (Referee 2) · 2023-7-2

Strengths

New ideas on entries in the holographic dictionary

Weaknesses

Wording is sometimes not clear and it was tedious to understand the set-up

Report

The author investigates a new set of holographic pairs: the volume of spacetime would be associated to the probability measure constrained with quantum dynamics and the Einstein eq. obtained through a semi-classical limit of generator equations describing the evolution of probability of joint distributions.

The ideas are interesting and new, however it sometimes lack of clarity. I am not sure I understood the gravity set-up the author is using. see my clarification questions.

1. I'm not sure I understand the picture of the spacetime in JT gravity, it said that: ``spacetime as factorized into a rigid two-dimensional space and a compactified sphere''. There is only two dimensions in JT gravity and the dilaton. To which sphere the author is referring too?

2. Does a potential non-trivial topology for AdS would change something about the arguments?

3. The author discusses flat JTs. It seems that this corresponds in the draft to the cosmological constant to zero (though it is put to 1 in section 2.).
However flat JT doesn't only correspond to Lambda to zero but one needs to shift the dilaton (not doing this shift will yield a theory without black holes). See for instance: 2112.14609. Which set-up does the author use? Does the presence of black holes modify the results?
Other small questions on section 2.1 what $n$ is normal to? Why Dirichlet boundary conditons are more natural? What the stress tensor does not depend on the boundary metric? I don't understand how the equations (11) - (12) are obtained from (10) - is a gauge chosen for the metric?

For the introduction:
4. In eq. (4), it would be good to add a sentence of what is $T_1$? Why is it important to associated this extra variable?
Same remark for (5), (6), it would be nice to describe in words what all the symbols refer too, in particular for non experts.

5. Where is the matter stress tensor in eq. (6) while it appears in (2) ?

For section 3. I appreciate the review the author has made.

For section 4: I cannot pronounce myself as I'm not sure to have understand which theory the author was referring too.

General questions:
6. Could the author comment on how this scenario will be uplift to higher dimensions. Are these considerations purely two dimensional?

  • validity: good
  • significance: high
  • originality: high
  • clarity: low
  • formatting: -
  • grammar: -

Author:  Josephine Suh  on 2023-08-01  [id 3861]

(in reply to Report 2 on 2023-07-02)
Category:
answer to question

We thank the referee for helpful questions, and hope our answers below can be satisfactory.

1. I'm not sure I understand the picture of the spacetime in JT gravity, it said that: spacetime as factorized into a rigid two-dimensional space and a compactified sphere''. There is only two dimensions in JT gravity and the dilaton. To which sphere the author is referring too?

Dilatonic theories on an $AdS_2$ background often arise by dimensional reduction of an $AdS_2 \times X$ geometry. (See e.g. reference 1402.6334.) For example, $X$ can be $S^2$ and after dimensional reduction, the the value of the dilaton defined as in our manuscript corresponds to the area of the compact two-sphere.

2. Does a potential non-trivial topology for AdS would change something about the arguments?

Our arguments work to all orders in a perturbative expansion in $N$ of SYK, or $\gamma$ in our setup. To this accuracy, one has a background spacetime in Lorentzian signature which is $AdS_2$ with two boundaries. AdS spacetimes with non-trivial topologies enter Euclidean path integrals to subleading order in an expansion in $e^N$. These are relevant in situations where one is calculating quantities that are sensitive to such corrections. In our setup we are not asking such questions.

3. The author discusses flat JTs. It seems that this corresponds in the draft to the cosmological constant to zero (though it is put to 1 in section 2.). However flat JT doesn't only correspond to Lambda to zero but one needs to shift the dilaton (not doing this shift will yield a theory without black holes). See for instance: 2112.14609. Which set-up does the author use? Does the presence of black holes modify the results?

We do not expect to have a black hole when we take the flat-space limit in our setup. In particular, our limit of flat space involves taking the temperature to be zero, see eq. (63).

Other small questions on section 2.1 what n is normal to?

$n$ is a normal to the level curve of the dilaton.

Why Dirichlet boundary conditons are more natural?

Dirichlet boundary conditions ensure that the JT action reduces to the Schwarzian action, which captures the low-energy dynamics of the SYK model.

What the stress tensor does not depend on the boundary metric?

The stress tensor can exist as an arbitrary source for the dilaton field in the bulk spacetime.

I don't understand how the equations (11) - (12) are obtained from (10) - is a gauge chosen for the metric?

There is no gauge chosen for the metric. However, to derive the formulas, it is convenient to work in null coordinates. We did not include the derivation in an Appendix because (11)-(12) are not important to the main content of the paper. It merely serves as motivation.

For the introduction: 4. In eq. (4), it would be good to add a sentence of what is T1? Why is it important to associated this extra variable? Same remark for (5), (6), it would be nice to describe in words what all the symbols refer too, in particular for non experts.

We thank the referee's suggestion. We will include a description of the $T_i$'s as the times of the projections $P_i$.

5. Where is the matter stress tensor in eq. (6) while it appears in (2) ?

In (2), we are considering the most general equation of motion resulting from (1). In Eq. (6), we are working in JT gravity without matter. We may include a clarification on this point.

General questions: 6. Could the author comment on how this scenario will be uplift to higher dimensions. Are these considerations purely two dimensional?

As we describe in the second-to-last paragraph in the introduction, we conjecture that the general identification of spacetime as target space and volume measure as probability measure, and Einstein's equations as a generator equation, will hold quite generally in general relativity, and not just in two dimensions.

However, the technical derivation of the above results in JT gravity certainly do depend on the details and solvable nature of the theory. In ongoing and future work, we hope to give proposals and theoretical machinery for calculating the relevant quantities we have identified in this paper (i.e. joint quantum distributions) in higher-dimensional gravitational theories.

---

## Round 3 · Author Response

I have composed replies to both of the referee's reports, answering their questions, and have resubmitted a revised version of the manuscript that includes additional exposition as appropriate.
Best regards,
Josephine Suh

---

## Round 3 · List of Changes

The new Conclusion and Discussion section includes an exposition of
1. the role that the microscopic SYK model plays in our discussion
2. how the bulk spacetime AdS_2 emerges and our new proposal regarding the emergent spacetime
2. the role of Dirichlet b.c. and negative cosmological constant in the JT action
3. relation to previous research deriving Einstein's equations from entanglement entropy.

---

## Editorial Decision

published